# A broadly neutralizing antibody protects Syrian hamsters against SARS-CoV-2 Omicron challenge

Biao Zhou[1,2,12], Runhong Zhou[1,2,12], Bingjie Tang [3,12], Jasper Fuk-Woo Chan [2,4,5,6,7,12], Mengxiao Luo[1,2,12], Qiaoli Peng[1,2,8,12], Shuofeng Yuan[2,4,5,6,7,12], Hang Liu [3,12], Bobo Wing-Yee Mok[2,4,5], Bohao Chen[1,2], Pui Wang[2,4,5], Vincent Kwok-Man Poon [2], Hin Chu [2,4,5], Chris Chung-Sing Chan [2], Jessica Oi-Ling Tsang[2], Chris Chun-Yiu Chan[2], Ka-Kit Au[1,2], Hiu-On Man[1,2], Lu Lu[2], Kelvin Kai-Wang To [2,4,5,6,7], Honglin Chen [2,4,5,6], Kwok-Yung Yuen [2,4,5,6,7,9], Shangyu Dang [3,10,11✉] & Zhiwei Chen [1,2,4,5,6✉]

The strikingly high transmissibility and antibody evasion of SARS-CoV-2 Omicron variants have posed great challenges to the efficacy of current vaccines and antibody immunotherapy. Here, we screen 34 BNT162b2-vaccinees and isolate a public broadly neutralizing antibody ZCB11 derived from the IGHV1-58 family. ZCB11 targets viral receptor-binding domain specifically and neutralizes all SARS-CoV-2 variants of concern, especially with great potency against authentic Omicron and Delta variants. Pseudovirus-based mapping of 57 naturally occurred spike mutations or deletions reveals that S371L results in 11-fold neutralization resistance, but it is rescued by compensating mutations in Omicron variants. Cryo-EM analysis demonstrates that ZCB11 heavy chain predominantly interacts with Omicron spike trimer with receptor-binding domain in up conformation blocking ACE2 binding. In addition, prophylactic or therapeutic ZCB11 administration protects lung infection against Omicron viral challenge in golden Syrian hamsters. These results suggest that vaccine-induced ZCB11 is a promising broadly neutralizing antibody for biomedical interventions against pandemic SARS-CoV-2.

[1] AIDS Institute, Li Ka Shing Faculty of Medicine, The University of Hong Kong, Pokfulam, Hong Kong Special Administrative Region, People's Republic of China. [2] Department of Microbiology, Li Ka Shing Faculty of Medicine, The University of Hong Kong, Pokfulam, Hong Kong Special Administrative Region, People's Republic of China. [3] Division of Life Science, The Hong Kong University of Science and Technology, Clear Water Bay, Kowloon, Hong Kong Special Administrative Region, People's Republic of China. [4] State Key Laboratory of Emerging Infectious Diseases, The University of Hong Kong, Pokfulam, Hong Kong Special Administrative Region, People's Republic of China. [5] Centre for Virology, Vaccinology and Therapeutics, Health@InnoHK, The University of Hong Kong, Hong Kong, Hong Kong Special Administrative Region, People's Republic of China. [6] Department of Clinical Microbiology and Infection Control, The University of Hong Kong-Shenzhen Hospital, Shenzhen, Guangdong, People's Republic of China. [7] Department of Microbiology, Queen Mary Hospital, Pokfulam, Hong Kong Special Administrative Region, People's Republic of China. [8] National Clinical Research Center for Infectious Diseases, HKU-AIDS Institute Shenzhen Research laboratory, The Third People's Hospital of Shenzhen, The Second Affiliated Hospital of Southern University of Science and Technology, Shenzhen, Guangdong, People's Republic of China. [9] Academician Workstation of Hainan Province and Hainan Medical University-The University of Hong Kong Joint Laboratory of Tropical Infectious Diseases, The University of Hong Kong, Pokfulam, Hong Kong Special Administrative Region, People's Republic of China. [10] Southern Marine Science and Engineering Guangdong Laboratory (Guangzhou), Guangzhou, People's Republic of China. [11] Center of Systems Biology and Human Health, Hong Kong University of Science and Technology, Clear Water Bay, Kowloon, Hong Kong Special Administrative Region, People's Republic of China. [12] These authors contributed equally: Biao Zhou, Runhong Zhou, Bingjie Tang, Jasper Fuk-Woo Chan, Mengxiao Luo, Qiaoli Peng, Shuofeng Yuan, Hang Liu. ✉email: sdang@ust.hk; zchenai@hku.hk

After two years of the COVID-19 pandemic, the highly transmissible SARS-CoV-2 and its variant of concerns (VOCs) have resulted in more than 493 million infections with 6.15 million deaths globally by March 31, 2022 (https://coronavirus.jhu.edu/map.html). During this period, various types of COVID-19 vaccines have been quickly developed to control the pandemic with over 11 billion doses administered in the world. Although the extensive implementation of vaccination has significantly reduced the rates of hospitalization, disease severity, and death[1–5], current vaccines do not confer complete or durable prevention of upper airway transmission of SARS-CoV-2. The numbers of vaccine-breakthrough infections and re-infections, therefore, have been continuously increasing[6–8]. The pandemic situation has been complicated by repeated emergence of new VOCs, including Alpha (B.1.1.7), Beta (B.1.351), Gamma (P.1), Delta (B.1.617.2), and Omicron (B.1.1.529)[9,10], and waning of vaccine-induced immune responses, together with relaxed preventive masking and social distancing[11–13].

After the World Health Organization (WHO) designated the Omicron as a VOC on November 26, 2021, this variant has been quickly found in over 170 countries and replaced the Delta VOC within a short period, becoming the dominant VOC in many places in the South Africa, European countries, and the United States[14,15]. The rapid global spread of the Omicron VOC has been associated with vaccine-breakthrough infections and re-infections[16,17]. Like previous findings that the Beta VOC compromised vaccine-induced neutralizing antibody (NAb)[12,18,19], the Omicron VOC has resulted in even worse NAb evasion due to more than 30 alarming mutations in SARS-CoV-2 spike glycoprotein[20–23]. This situation continues to evolve with increased global spreading of Omicron variants including BA.1, BA.1.1, and BA.2[24]. Considering that current NAb combination for clinical immunotherapy showed greatly reduced activities[21,25], it is necessary to discover broadly neutralizing antibodies (bNAbs) that fully cover the diverse VOCs of SARS-CoV-2.

NAbs target epitopes in SARS-CoV-2 receptor-binding domain (RBD), N-terminal domain (NTD), and quaternary regions. With RBD as the primary target, four classes of NAbs have been described based on antibody competition and structural studies[20,21,26,27]. Accordingly, class I antibodies bind to the tip of RBD in the up conformation whereas class II antibodies recognize the ridge of RBD either in up or down configuration. Both classes block ACE2-binding site directly. Class III antibodies interact with the RBD core engaging the N343 glycan without direct overlapping with ACE2-binding site. Class IV antibodies target a cryptic and conserved epitope, which is only accessible when at least one RBD is in the up conformation. Omicron variant with 15 RBD mutations, however, can escape most NAbs through various mechanisms. For example, class I NAbs (e.g., B1-182.1, S2E12, CB6) were through a steric clash (Q493R) and removal of key contacts (K417N and Y505H). Class II NAbs (e.g., LY-COV555, A19-46.1, and ZB8) were caused by a steric clash (E484A or Q493R) or by the S371L/S373P/S375F alterations that restrict the RBD-up conformation. Class III NAbs (e.g., A19-61.1, COV2-2130, S309) were affected by mutations surrounding the mutation-free RBD surfaces, but LY-CoV1404 was not affected[27].

In this study, we seek to search for vaccine-induced bNAbs with potency capable of overcoming resistance conferred by circulating SARS-CoV-2 Omicron variants. The mRNA vaccine-induced public bNAb ZCB11 derived from the IGHV1-58 family potently neutralizes all SARS-CoV-2 VOCs tested by interacting with RBD predominantly via the heavy chain. In addition, prophylactic and therapeutic ZCB11 administration protects lung infection against Omicron viral challenge in golden Syrian hamsters, displaying promising biomedical interventions against pandemic SARS-CoV-2.

## Results

### Identification of a BNT162b2-vaccinee who developed bNAbs.
To isolate potent bNAbs against currently circulating SARS-CoV-2 VOCs, we searched for individuals, who had developed potent bNAbs among a Hong Kong cohort of 34 vaccinees, around average 30.7 days (range, 7–47 days) after their second dose of the BNT162b2 vaccination (BioNTech-Pfizer) (Supplementary Table 1)[13]. All subjects developed NAbs against the pseudotyped SARS-CoV-2 wildtype (WT, D614G) (Fig. 1a). To seek for vaccinees with bNAb, we then tested their neutralizing activities against the full panel of pseudotyped SARS-CoV-2 VOCs including Alpha (B.1.1.7), Beta (B.1.351), Gamma (P.1), Delta (B.1.617.2) and Omicron BA.1 (B.1.1.529) (Fig. 1b–f). Only two study subjects (2/34), BNT162b2-26 and BNT162b2-55, were considered as candidate vaccinees who harbored bNAbs with $IC_{90}$ or $IC_{50}$ values higher than the mean titers of all VOCs tested in the cohort. Interestingly, BNT162b2-26 displayed greatly high bNAb titers against the Beta and Delta variants (Fig. 1c and e, Supplementary Table 2), the known most resistant VOC and the dominant VOC, respectively, before the Omicron variants[28,29]. After measuring binding antibodies to spike protein (Fig. 1g), we calculated the neutralizing potency index as previously described[30]. We found that Omicron BA.1 resulted in the highest reduction of the mean neutralizing potency index as compared with other VOCs. BNT162b2-26, however, displayed neutralizing potency index scores consistently higher than the mean ones against all VOCs tested (Fig. 1h). We, therefore, chose this vaccinee for subsequent search of bNAbs.

### Isolation of NAbs against SARS-CoV-2 from BNT162b2-26.
With vaccinee informed consent, we obtained another blood sample donated by BNT162b2-26 at day 130 after his second vaccination. Fresh PBMCs from BNT162b2-26 were stained for antigen-specific memory B cells (CD19, CD27, IgG) using the 6×His-tagged SARS-CoV-2 WT spike as the bait as previously described[31]. Spike-specific memory B cells were found in BNT162b2-26 but not in the healthy donor (HD) control (Supplementary Fig. 1) and were sorted into each well with a single B cell for antibody gene amplification. After antibody gene sequencing, we recovered 14-paired heavy chain and light chain for antibody IgG1 engineering. Seven of these 14-paired antibodies in culture supernatants including ZCB3, ZCB8, ZCB9, ZCB11, ZCC10, ZCD3, ZCD4 showed positive responses to WT spike by ELISA 48 h post transient transfection (Supplementary Fig. 2a). Five of these seven spike-reactive antibodies including ZCB3, ZCD4, ZCB11, ZCC10, and ZCD3 targeted spike S1 subunit (Supplementary Fig. 2b), whereas ZCB8 was weak but S2-specific by both ELISA and Western blot experiments (Supplementary Fig. 2c and g). Moreover, among these five S1-reactive antibodies, only ZCD4 was not RBD-specific (Supplementary Fig. 2d) and none of them interacted with NTD (Supplementary Fig. 2e, Supplementary Table 3). Four RBD-specific ZCB3, ZCB11, ZCC10, and ZCD3 showed neutralizing activities against WT by the pseudovirus neutralization assay (Supplementary Fig. 2f). These results demonstrated that RBD-specific monoclonal NAbs were successfully obtained from memory B cells of BNT162b2-26.

Notably, besides the previously published control ZB8[31], ZCB11 had the strongest binding capability to both RBD and spike with the same $EC_{50}$ values of 20 ng/ml by ELISA (Fig. 2a, b, Supplementary Table 4). Moreover, the binding dynamics of ZCB11 to SARS-CoV-2 RBD was determined by the surface plasmon resonance (SPR). We found that ZCB11 exhibited an equilibrium dissociation constant (KD) value of $5.75 \times 10^{-11}$ M, suggesting an RBD-specific high-binding affinity (Supplementary

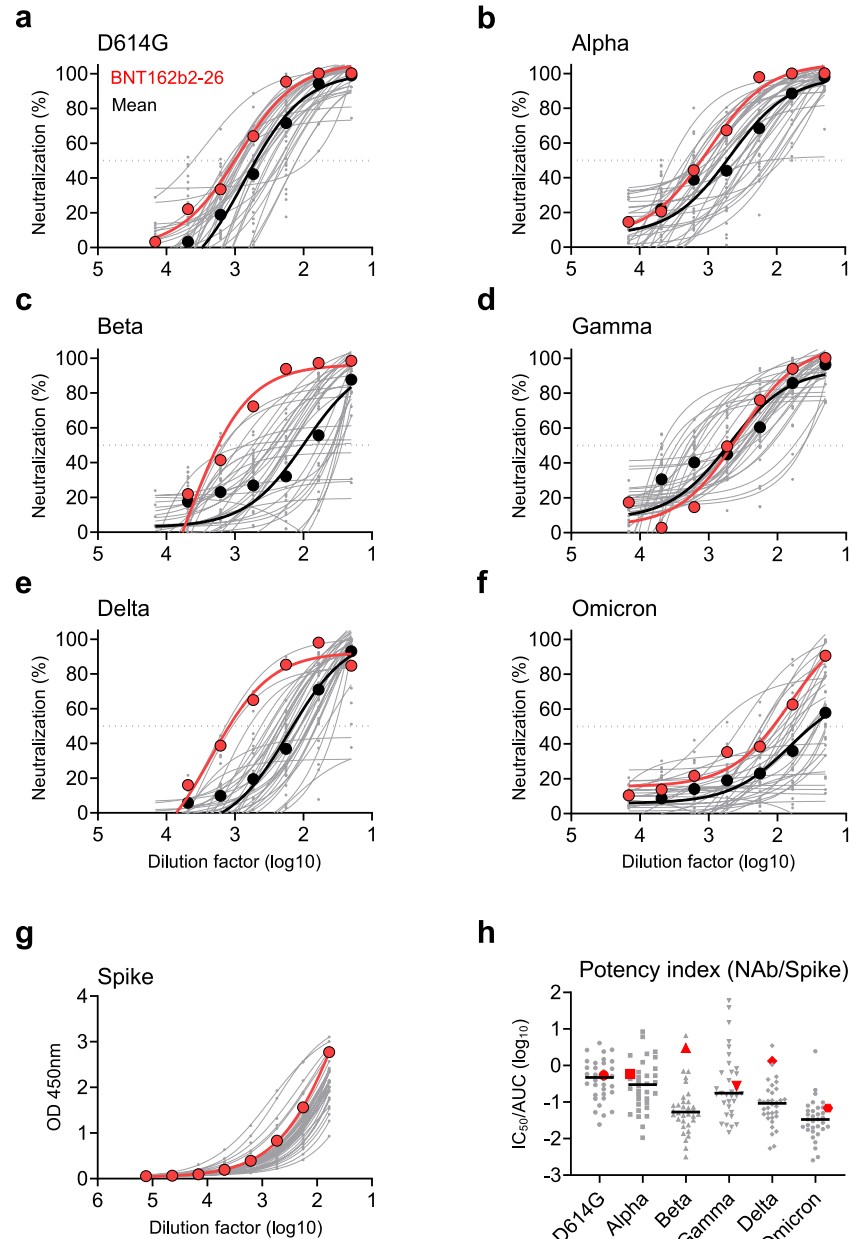

**Fig. 1 Identification of a vaccinee who developed bNAbs.** Plasma samples derived from 34 BNT162b2-vaccinees were tested at average 30.7 days (range 7-47 days) after second vaccination. **a–f** Serially diluted plasma samples were subjected to neutralization assay against the pseudotyped SARS-CoV-2 WT (**a**) and five VOCs (**b–f**), respectively. The neutralizing curve of the BNT162b2-26 vaccinee (red) was compared with the mean curve of all vaccinees tested (dark black). **g** Binding activity of spike-specific plasma IgG was determined by ELISA at serial dilutions. The binding curve of the BNT162b2-26 vaccinee was presented in red. **h** The neutralization antibody potency index was defined by the ratio of $IC_{50}$/AUC of anti-spike IgG in BNT162b2-vaccinees. Neutralizing $IC_{50}$ values represented plasma dilution required to achieve 50% virus neutralization. The area under curve (AUC) represented the total peak area was calculated from ELISA OD values. Each symbol represented an individual vaccinee with a line indicating the median of each group. The BNT162b2-26 vaccinee who developed bNAbs was presented as red symbols. The experiments were performed in parallel. Source data are provided as a Source Data file. bNAbs broadly neutralizing antibodies, WT wildtype, VOC variant of concern, ELISA enzyme-linked immunosorbent assay, $IC_{50}$ half inhibitory concentration, AUC area under curve, OD optical density.

Fig. 2h and Supplementary Table 5). In subsequent quantitative neutralization analysis against WT, we found that two of these four NAbs, ZCB3 and ZCB11, showed high neutralization potency with $IC_{50}$ values below 100 ng/mL (Fig. 2c, Supplementary Table 6). Sequence analysis revealed that ZCB3, ZCC10, and ZCD3 utilized IGHV3-53/3-66 heavy chain, whereas their paired light chains had distinct IGKV1-9, IGKV3-20, and IGKV1-27, respectively (Supplementary Table 7). In contrast, ZCB11 utilized IGHV1-58 heavy chain and IGKV3-20 light chain. Our four new

NAbs were all considered as public antibodies characterized by a IGHV3-53/3-66 heavy chain with 10–12 residues in the CDR3 region or an IGHV1-58 heavy chain with 15–17 residues in CDR3 region as previously reported by others[12,32,33]. These results demonstrated that we isolated four public NAbs from BNT162b2-26 after two standard doses of vaccination.

**Antibody neutralization of SARS-CoV-2 VOCs.** To understand the breadth of these four newly cloned public RBD-specific NAbs,

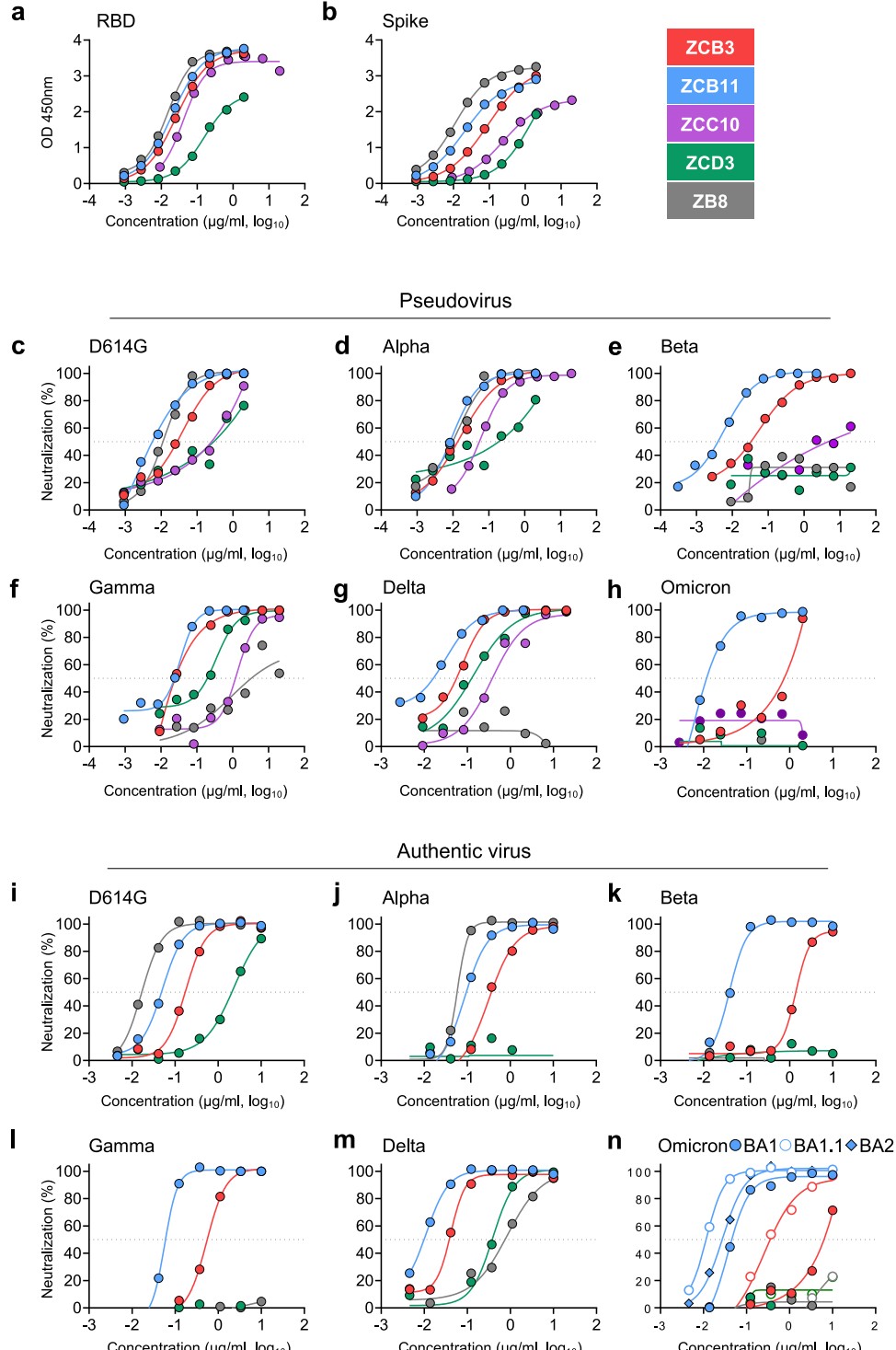

**Fig. 2 Comparison of bNAbs isolated from the BNT162b2-26 vaccinee. a**, **b** RBD- and spike-specific binding activities of four newly cloned NAbs including ZCB3, ZCB11, ZCC10, and ZCD3 were determined by ELISA at serial dilutions. A known NAb ZB8 was included as a control. **c**–**h** Neutralizing activities of ZCB3, ZCB11, ZCC10, and ZCD3 were determined against six pseudotyped SARS-CoV-2 variants, including D614G (WT), Alpha, Beta, Gamma, Delta, and Omicron as compared with ZB8. **i**–**n** Neutralizing activities of ZCB3, ZCB11, ZCC10, and ZCD3 were determined against the same but authentic SARS-CoV-2 variants, including Omicron BA.1, BA.1.1, and BA.2 as compared with ZB8. The color coding was consistently used in **a**–**n**. The dashed line in each graph indicated 50% neutralization. The experiments were performed in at least duplicates. Source data are provided as a Source Data file. RBD receptor-binding domain, ELISA enzyme-linked immunosorbent assay, bNAbs broadly neutralizing antibodies, WT wildtype.

we performed SARS-CoV-2 neutralization assays using both pseudoviruses and authentic VOC isolates, including Alpha, Beta, Gamma, Delta, and Omicron variants (Figs. 2c–h and 2i–n). ZB8, a known RBD-specific class II NAb[31], was included as a positive control. Testing pseudoviruses in 293T-ACE2 cells, we found that ZCB11 was the best bNAb that neutralized all VOCs potently, including the highly transmissible and prevalent Omicron BA.1[21], with $IC_{50}$ values of elite 6 ng/mL for Alpha, Beta, and Omicron BA.1 variants and 30 ng/mL for Gamma and Delta variants (Fig. 2c–h, Supplementary Table 6). ZCB3 was the second-best bNAb and neutralized Alpha, Beta, Gamma, and Delta variants potently, but not the Omicron BA.1. ZCC10 and ZCD3 neutralized Alpha, Gamma, and Delta variants at relative low potency, but lost neutralization against Beta and Omicron BA.1 totally. Importantly, testing authentic VOC viruses in Vero-E6-TMPRSS2 cells, we consistently found that ZCB11 was the most potent bNAb, followed by ZCB3 (Fig. 2i–n). The $IC_{50}$ values of ZCB11 for neutralizing Alpha, Beta, Gamma, Delta, Omicron BA.1, Omicron BA.1.1, and Omicron BA.2 variants were 85.1, 39.9, 56.9, 11.2, 36.8, 11.7, and 27.7 ng/mL respectively, which were comparable to or better than the $IC_{50}$ value of 51 ng/ml for neutralizing the WT (Supplementary Table 6). ZCB3 was about 10-fold less potent than ZCB11 for neutralizing Beta and Omicron variants. Notably, the potency of ZCB11 in the pseudovirus assay was higher than that in the authentic virus assay, which was probably related to different target cells and assays used. ZB8 showed unmeasurable and weak neutralization against Delta pseudovirus and Delta authentic virus, respectively. ZCC10 and ZCD3 showed weak and unmeasurable neutralization against Gamma pseudovirus and Gamma authentic virus, respectively. These results demonstrated that ZCB11 functioned as an elite bNAb that potently neutralized all circulating SARS-CoV-2 VOCs in vitro.

**Naturally occurred mutations or deletions conferring antibody neutralization resistance.** Since Omicron variants escaped from majority of known RBD-specific NAbs[20,21,26], we sought to determine possible mutations or deletions responsible for antibody resistance for ZCB3 and ZCB11 as compared with the control ZB8. We first constructed and tested a large panel of pseudoviruses carrying individual mutations or deletions found in Omicron BA.1 as compared with those previously found in Alpha, Beta, Gamma, and Delta variants (Fig. 3a and Supplementary Fig. 3). For the control ZB8, a class II NAb, we consistently found that the E484 was essential for its neutralization activity. E484K in Beta, E484Q in Kappa, and E484A in Omicron were responsible for the significant ZB8 resistance, followed by Q493R for about 10-fold resistance. For ZCB3, none of single mutations or deletions tested conferred resistance for equal to or more than 10-fold. Omicron BA.1 and Q493R reduced neutralization potency of 13-fold and 4-fold, respectively. For ZCB11, S371L of Omicron BA.1 showed 11-fold resistance (Fig. 3a). Moreover, individual Q493R, Y505H, T547K, and Q954H of Omicron BA.1 exhibited about 6-fold resistance. Unexpectedly, when all these and other mutations combined in Omicron BA.1, they did not confer significant resistance at all. Subsequently, we performed antibody competition by the SPR. ZCB11 exhibited as a strong competitor for WT RBD binding against either ZCB3 or ZB8, respectively, (Supplementary Fig. 4a–d), suggesting overlapped antibody binding epitopes in RBD between them. Considering that some ultrapotent public NAbs were also derived from IGHV1-58 family[32,34,35], we sought to make a direct comparison by generating S2E12, 2C08, B1-182.1, and COV2-2072. We found that ZCB11 displayed a strong competition with these public NAbs for binding to SARS-CoV-2 RBD (Supplementary Fig. 4e to 4h), but only ZCB11 exhibited the fast-on/slow-off kinetics by the SPR analysis (Fig. 3b). Moreover, ZCB11 displayed

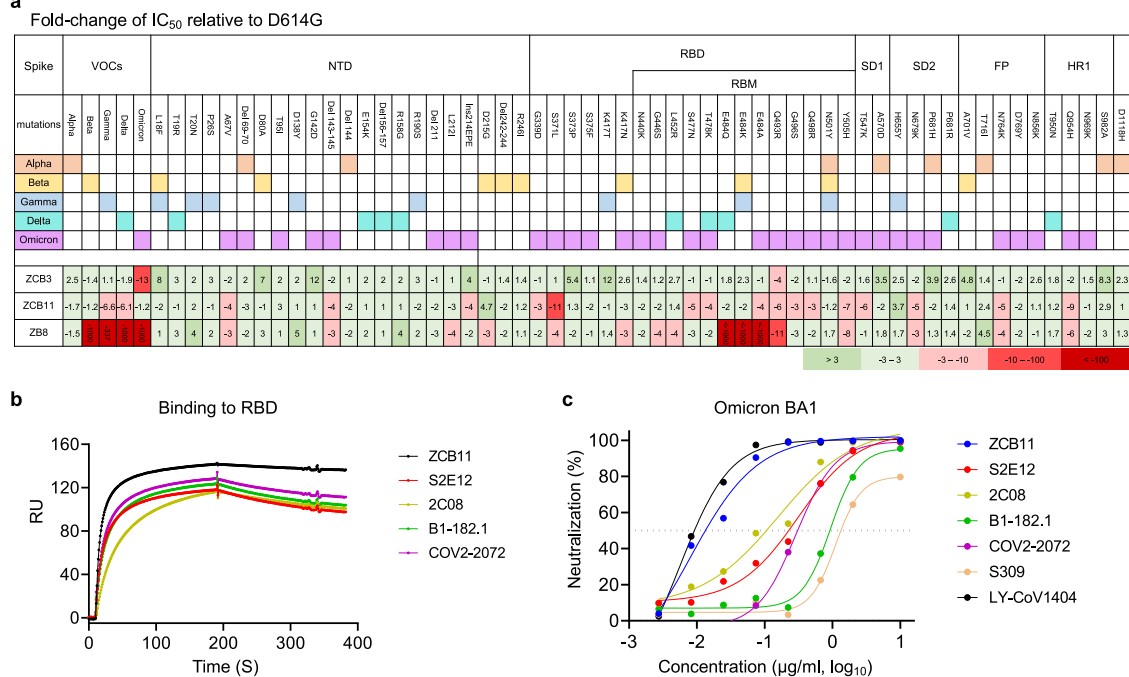

**Fig. 3 ZCB11 resistance to natural mutations in SARS-CoV-2 VOCs and comparison with four public antibodies derived from IGHV1-58 family. a** Fold change of $IC_{50}$ values relative to WT was determined by pseudoviruses carrying individual mutations or deletion against ZCB3 and ZCB11 as compared with ZB8. **b** Binding kinetics of ZCB11 to SARS-CoV-2 RBD was determined by SPR as compared with four public antibodies derived from IGHV1-58 family. The color coding indicates each individual antibody. **c** Neutralization of pseudotyped Omicron BA.1 by ZCB11 and four public antibodies as compared with two therapeutic antibodies (S309 and LY-CoV1404). The experiments were performed in duplicates. Source data are provided as a Source Data file. VOC variant of concern, $IC_{50}$ half inhibitory concentration, WT wildtype, RBD receptor-binding domain, SPR surface plasmon resonance.

more potent neutralizing activity than S2E12, 2C08, B1-182.1, COV2-2072, and S309 against the pseudotyped Omicron BA.1 variant but was slightly weaker than LY-CoV1404 (Fig. 3c). These results demonstrated that ZCB11 was an elite neutralizer with relatively higher affinity, potency, and breadth than similar public NAbs tested in parallel.

**Structure of SARS-CoV-2 Omicron Spike with ZCB11 Fab bound.** To understand the mechanism of SARS-CoV-2 Omicron

variants potently neutralized by ZCB11, we determined the structure of Omicron spike trimer in complex with ZCB11 Fab (spike-Fab) using single-particle cryo-electron microscopy (Cryo-EM) at 3.4 Å (Fig. 4, Supplementary Fig. 5 and 6a and 6b, Supplementary Table 8). The spike-Fab complex showed three major states in the RBD region. In one state, all three RBDs presented "up" conformation (3 u). In other two states, only 2 RBDs exhibited "up" conformation and the remaining RBD showed "down" conformation (2u1d), despite a slight shift of the RBD

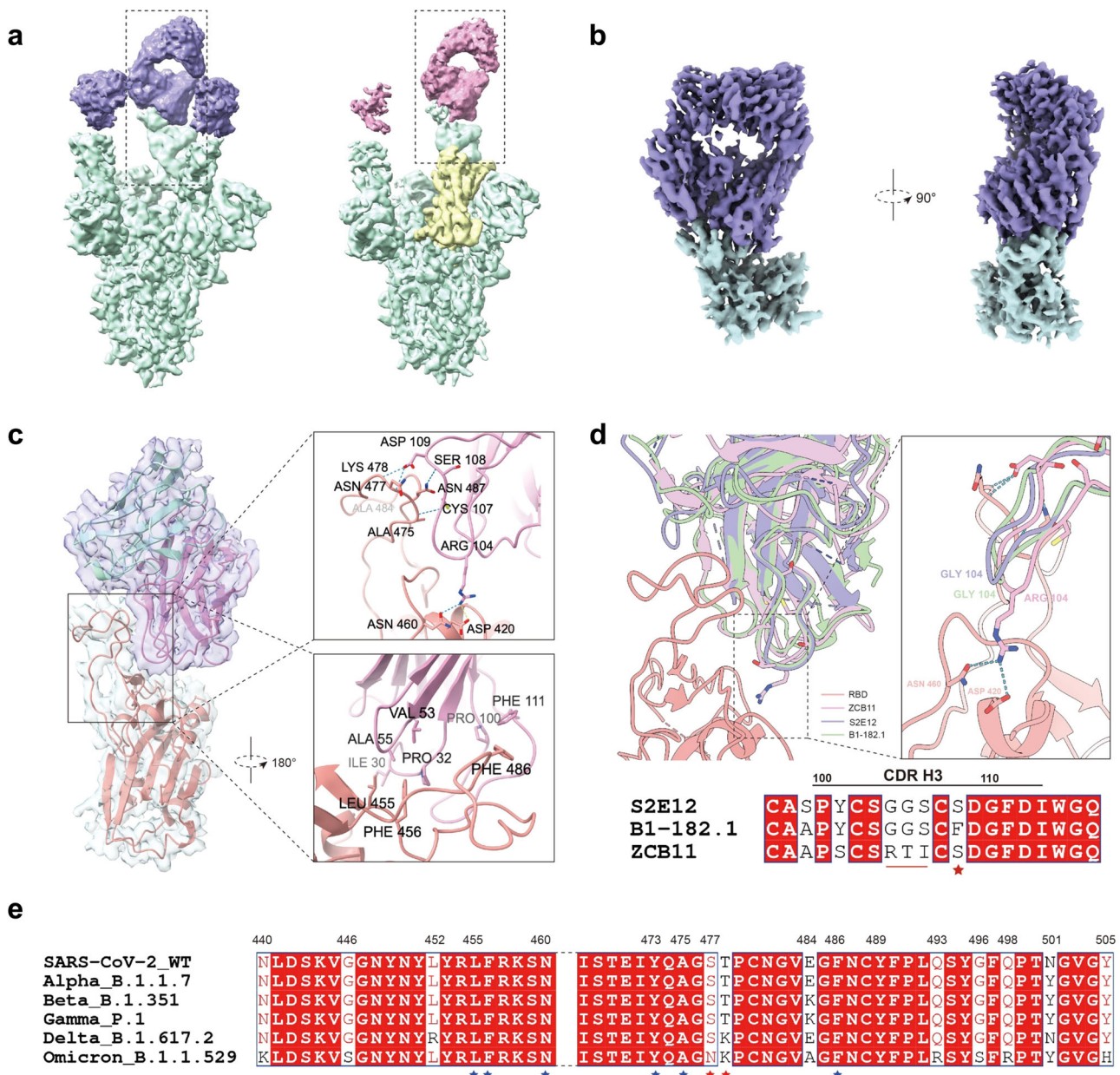

**Fig. 4 Cryo-EM structure of SARS-CoV-2 Omicron BA.1 Spike with ZCB11 Fab bound. a** Cryo-EM density map of spike trimer in complex with ZCB11 Fab. Two of three different states (3 u and 2u1d) are shown. Spike trimer is color-coded in green and Fab is in pink and purple in two states, respectively. Down RBD is color-coded in yellow. **b** Cryo-EM density map of RBD-Fab complex shown in two different views. Fab is color-coded in purple and RBD is in green. **c** Interaction between ZCB11 Fab and RBD. ZCB11 is shown as heavy chain (pink) and light chain (blue). Two interfaces are zoomed to show key residues responsible for the interaction. Hydrogen bonds and salt bridges are indicated as blue and yellow dashed lines, respectively. Key contact residues are also indicated. **d** Binding pattern of ZCB11 (purple) is compared with that of B1-182.1 (green, PDB: 7MLZ) or S2E12 (violet, PDB: 7K45). Sequence alignment shows the CDR H3 region including the unique RTI motif (underlined) in ZCB11. The red star highlights the S108 residue. **e** Sequence alignment of RBMs from different SARS-CoV-2 VOCs. Mutations and key residues are marked with numbers. Conserved key interface residues and mutant residues are labeled with blue and red stars, respectively. EM electron microscope, Fab antibody-binding fragment, RBD receptor-binding domain, CDR complementarity-determining region, RBM receptor-binding motif, VOCs variants of concern.

region between two states (Fig. 4a, Supplementary Fig. 6). In all three states, only RBD in the "up" conformation interacted with ZCB11 Fab (Fig. 4b), suggesting the property of class I NAbs[27,36,37]. The "down" RBD did not bind to the Fab because its epitope was blocked by the neighboring "up" RBD (Fig. 4a, Supplementary Fig. 6c), similar to previous findings on some class I and class IV antibodies[36]. After 3D classification without alignments focusing on the RBD-Fab region, particles with clear RBD-Fab features were selected for subtraction and local refinement to further improve the local resolution of RBD-Fab (Supplementary Fig. 6). Structural analysis showed that the epitope in RBD interacting with ZCB11 partially overlapped with the ACE2-binding site (Supplementary Fig. 6d), suggesting that ZCB11 is an ACE2 blocker similar to other class I antibodies[27,36]. Further structural analysis revealed two binding interfaces between the ZCB11 heavy chain and RBM (Receptor-Binding Motif) (Fig. 4c), which is different from typical class I antibodies involving both heavy and light chains for binding[27]. Furthermore, the binding of ZCB11 to RBM was mainly driven by the first interface by forming tight interaction through strong hydrogen bonds, including those formed between N477, K478, N487, N460 of RBM and D109, S108, R104 in CDR H3 or S34 in CDR H1 of ZCB11 heavy chain, respectively (Fig. 4c). Another hydrogen bond formed between A475 of RBM and C107 in CDR H3 of ZCB11 heavy chain strengthened the interaction (Fig. 4c). In addition, the salt bridge, formed between R104 in CDR H3 of ZCB11 heavy chain and D420 of RBM further stabilized the binding of ZCB11 to RBM (Fig. 4c). Notably, the unique RTI motif at positions 104-106 in CDR H3 of ZCB11 contributed to direct interaction with RBM and this interface did not involve viral mutations accounting for resistance to other class I antibodies (Fig. 4d)[27]. The second relatively weaker interface between ZCB11 and RBM was stabilized by the hydrophobic interaction mediated by multiple residues from RBM (L455, F456, F486) and ZCB11 (I30, P32, V53, A55, P100, F111) (Fig. 4c). In addition, P32 of ZCB11 might facilitate its binding to RBM by forming CH/π interaction with F456 of RBM. Sequence analysis of RBM among all VOCs indicated that several highly conserved residues L455, F456, N460, Y473, and A475 played important roles in binding of ZCB11 (Fig. 4e). Interestingly, the S477N and T478K mutations in Omicron variant RBD likely contributed to the binding to ZCB11 (Fig. 4c, e), instead of the steric clash found in some class I antibodies for neutralization escape[27]. These results demonstrated that ZCB11 exhibited unique structural properties, enabling its potent and broad neutralization against Omicron variants and diverse VOCs.

**In vivo efficacy of ZCB11 against SARS-CoV-2 Omicron and Delta variants**. To determine the in vivo potency of ZCB11 against the dominant circulating VOCs, we conducted viral challenge experiments using the golden Syrian hamster COVID-19 model and used ZB8 for comparison[38]. Since ZB8 conferred nearly complete lung protection against SARS-CoV-2 WT intranasal challenge at 4.5 mg/kg as we previously described[39], we tested it in parallel with ZCB11 using the same dose according to our standard experimental procedure (Fig. 5a). One day prior viral challenge, three groups of hamsters ($n = 8$) received the intraperitoneal injection of ZCB11, ZB8, and PBS, respectively. Twenty-four hours later, half of the animals ($n = 4$) in each group were separated into subgroups and were challenged intranasally with $10^5$ PFU of SARS-CoV-2 Delta and Omicron BA.1, respectively. Animal body weight changes were measured daily until day 4 when all animals were sacrificed for endpoint analysis. For hamsters challenged with the Delta variant, we found that the infection caused around 10% body weight loss overtime in the

PBS and ZB8 pre-treatment groups (Fig. 5b). In contrast, transient and less than 4% body weight decrease was observed for hamsters pre-treated with ZCB11. Moreover, relatively lower subgenomic viral loads in both lung and NT as well as unmeasurable numbers of live infectious viruses (4–5 orders of magnitude drop) were achieved by ZCB11 than by ZB8 compared with the PBS group (Fig. 5c–f). For hamsters challenged with the Omicron BA.1 variant, no significant body weight loss was found in all three subgroups (Fig. 5g). Moreover, infected hamsters in the PBS group displayed relatively lower subgenomic viral loads and numbers of live infectious viruses in both lung and NT than corresponding animals infected with Delta, indicating relatively weaker pathogenicity caused by Omicron BA.1 than by Delta[40,41]. Significantly lower subgenomic viral loads in both lung and NT as well as unmeasurable numbers of live infectious viruses (3–4 orders of magnitude drop) were achieved only by ZCB11 compared with ZB8 and PBS groups (Fig. 5h–k). In addition, histopathological analysis further illustrated minimal lung lesions among hamsters pre-treated by ZCB11 against both Omicron BA.1 and Delta viral infections compared with control animals pre-treated with PBS displaying interstitial pneumonia with diffuse alveolar damage, alveolar septa thickness, perivascular inflammatory cell infiltration, edema of homogeneously pink materials as well as some hemorrhage foci (Fig. 5l). In contrast, hamsters pre-treated by ZB8 showed reduced lung pathology against Delta but not Omicron BA.1. These results demonstrated consistently that ZCB11 conferred significant protection against both Delta and Omicron BA.1 variants, whereas ZB8 exhibited only partial protection against Delta but not Omicron BA.1, in line with in vitro neutralizing activities of ZCB11 and ZB8 against live Delta and Omicron BA.1 variants, respectively (Fig. 2m, n).

To determine if ZCB11 could achieve therapeutic efficacy, we subsequently treated two groups of hamsters ($n = 5$) using the same dose of antibody (4.5 mg/kg) one and two days after the animals were challenged intranasally with $10^5$ PFU of SARS-CoV-2 Omicron BA.1 variant, respectively (Fig. 6a). Meantime, another group of hamsters ($n = 5$) were pre-treated with ZCB11, repeating the preventive experiment as described above. Consistent with findings in the previous experiment (Fig. 5h–k), ZCB11 pre-treatment resulted in significantly lower subgenomic viral loads as well as unmeasurable numbers of live infectious viruses in both lung and NT compared with the PBS group (Fig. 6b to 6e). Furthermore, ZCB11 post-treatment groups showed significant reductions of both subgenomic viral loads and infectious viral PFUs only in lungs (Fig. 6b, c). In the NT, however, both post-treatment groups could not reduce subgenomic viral loads significantly although they still achieved unmeasurable infectious PFU (Fig. 6d, e). Nevertheless, histopathological and immunofluorescence staining results showed consistently interstitial pneumonia and NP[+] cells mainly in lungs of hamsters treated with PBS (Figs. 5l, 6f and Supplementary Fig. 7). Reduced diffuse alveolar damage was found among hamsters pre-treated ($-1$ dpi) or early post-treated ($+1$ dpi) by ZCB11 against Omicron BA.1 infection compared with PBS and late post-treatment groups ($+2$ dpi) (Fig. 6f). Notably, PBS-treated hamsters in this experiment showed less diffuse alveolar damage and inflammatory cell infiltration but had similarly higher numbers of NP[+] cells (Supplementary Fig. 7). Since male hamsters were used in the first experiment but female hamsters in the second one, we believe that the observed discrepancy in lung damage was likely due to the gender difference as previously reported by others[42]. These results demonstrated that ZCB11 potently reduced lung damage and neutralized infectious Omicron BA.1 in both lung and NT but was unlikely able to significantly inhibit subgenomic viral loads in NT under post-exposure treatment settings like ZB8 as we previously described[39].

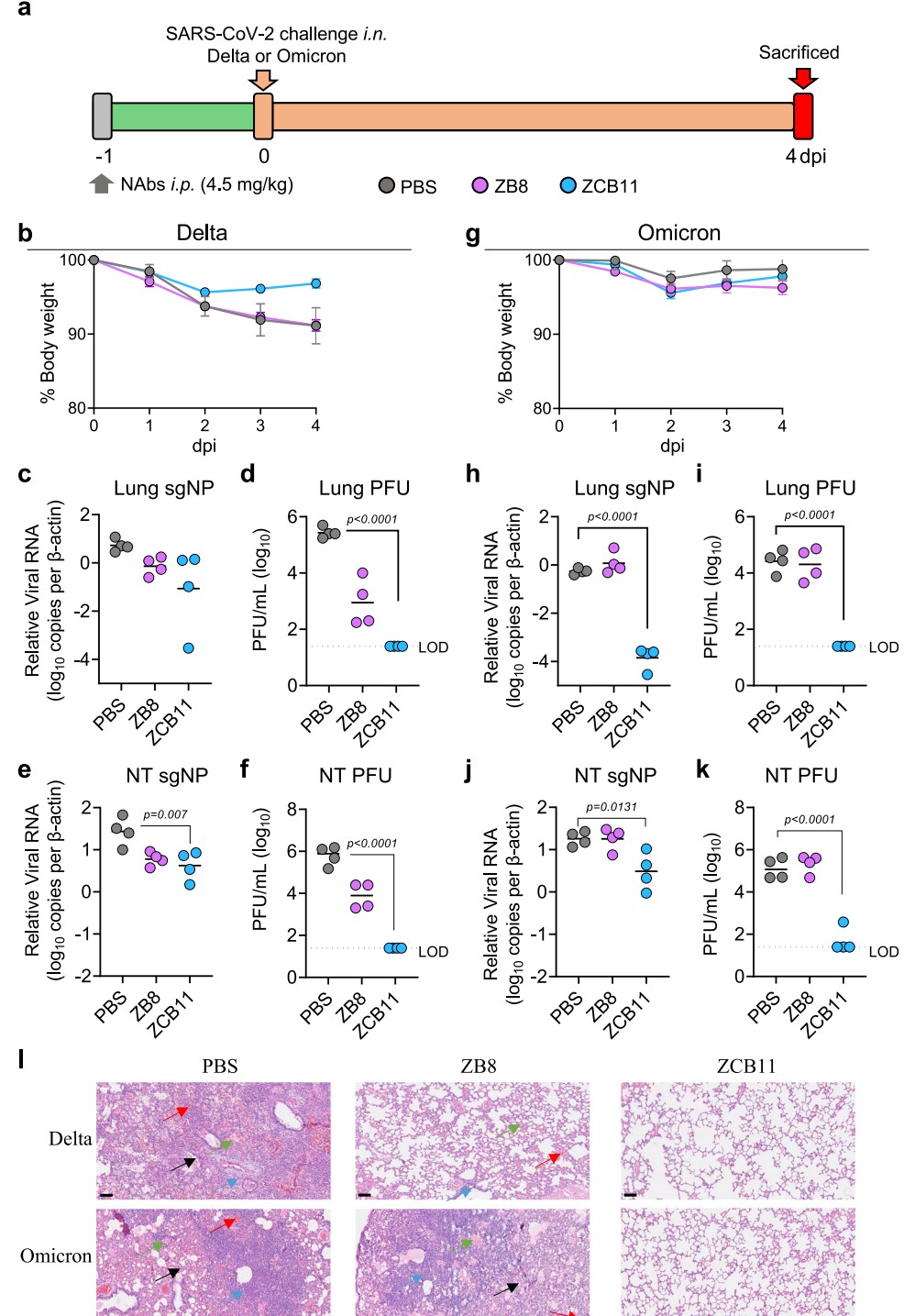

## Discussion

In this study, we showed that the standard two-dose BNT162b2 vaccination was able to induce spike-specific memory B cells, from which we successfully cloned the elite bNAb ZCB11 around 130 days post the second vaccination. We demonstrated that ZCB11 not only neutralized all authentic SARS-CoV-2 VOCs including three live pandemic Omicron variants at comparable high potency in vitro but also protected golden Syrian hamsters against the circulating Omicron BA.1 and Delta variants. We then demonstrated that ZCB11 only used its heavy chain to interact with Omicron spike trimer with RBD in up conformation, which revealed a unique structural basis in blocking ACE2 binding. In

particular, the unique interactive RTI motif in CDR H3 of ZCB11 has not been previously found with other RBD-specific class I NAbs derived from the same IGHV1-58 family. Our findings have significant implications for clinical development of ZCB11-based biomedical interventions against the pandemic SARS-CoV-2 VOCs.

ZCB11 overcomes naturally occurred spike mutations and deletions across current SARS-CoV-2 VOCs. Alpha variant with D614G and N501Y mutations enhanced RBD binding to human ACE2 receptor, transforming it into the most prevalent variant at the early stage of 2021[43]. N501Y confers partial resistance to RBD-specific class I NAb 910-30 but not NTD-specific NAb[21].

**Fig. 5 Prophylactic efficacy of ZCB11 against authentic SARS-CoV-2 Delta and Omicron BA.1 in golden Syrian hamsters as compared with ZB8.**
**a** Experimental schedule and color coding for different treatment groups. Three groups of male hamsters ($n = 8$) received a single intraperitoneal injection of PBS (gray), 4.5 mg/kg of ZB8 (purple) or 4.5 mg/kg of ZCB11 (blue) at one day before viral infection (−1 dpi). Twenty-four hours later (day 0), each group was divided into two subgroups for intranasal challenge with $10^5$ PFU live SARS-CoV-2 Delta and Omicron BA.1 variants, respectively. All animals were sacrificed on day 4 for final analysis. **b, g** Daily body weight change of each group ($n = 4$) was measured after viral infection. The data was shown as mean ± SEM. **c, e, h, j** The NP subgenomic RNA copy numbers (normalized by β-actin) in lung and nasal turbinate (NT) homogenates of each group ($n = 4$) were determined by a sensitive RT-PCR. The detection of viral load was performed in triplicates. **d, f, i, k** Live viral plaque assay was used to quantify the number of infectious viruses in lung and NT homogenates of each group ($n = 4$). Log10-transformed PFU per mL were shown for each group. The dash line indicates the limit of detection. **l** Representative histopathology of the lung tissues from pre-treated hamsters ($n = 4$ per group) after viral challenge. Tissue sections were stained with hematoxylin and eosin (H&E). In PBS-treated male hamsters, both Delta and Omicron could cause lung damages with alveolar septa thickening (black arrow), extensive inflammatory cell accumulation (blue arrow), homogeneously pink foci of edema (green arrow), and multifocal hemorrhage (red arrow). The scale bar represents 100 μm. PFU plaque-forming unit, SEM standard error of the mean, NP nucleocapsid protein, NT nasal turbinate, LOD limit of detection. Each symbol in **c–f** and **h–k** represents an individual hamster with a line indicating the mean of each group ($n = 4$). The color coding was consistently used in each graph. Statistics were generated using one-way ANOVA followed by Tukey's multiple comparisons test. P-values were shown on each graph where necessary. Source data (**b–j**) are provided as a Source Data file.

Subsequently, Beta, Gamma, and Delta variants displayed the most troublesome mutations, including K417N, E484K/Q/A, and N501Y, conferring high resistance to RBD-specific class I and class II NAbs[21,44–46]. E484K/Q/A led to almost complete loss of neutralization by potent RBD-specific class II NAbs such as LY-CoV555 and 2-15[21]. Due to antibody evasion, Delta variant, carrying L452R/T478K mutations, were found in more than 170 countries and accounted for 99% of newly confirmed cases before the Omicron variant[47–49]. After the emergence of the Omicron variant with more than 30 mutations in viral spike protein[50], the ongoing wave of COVID-19 pandemic has already been dominated by it over the Delta variant in many countries probably due to further antibody evasion to almost all current vaccines and NAbs including those approved for clinical use or emergency use[11,21,26,51]. N440K and G446S in Omicron conferred resistance to class III antibodies such as REGN10987 and 2–7[21]. G142D and del143-145 led to resistance to NTD-specific 4-18 and 5–7, whereas S371L conferred much broader resistance to RBD-specific class I, class III, and class IV NAbs including potent Brii-196, REGN10987, and Brii-198 in clinical development[21]. In this study, we consistently found that E484K/Q/A in Beta, Delta, and Omicron variants conferred strong resistance to our RBD-specific class II ZB8 NAb[39]. These resistant mutations, however, did not affect the potency of ZCB11 significantly. Although S371L found in Omicron displayed partial resistance (~11-fold) to ZCB11, similar amount of resistance was not observed against Omicron BA.1, BA.1.1, and BA.2 that all contained this mutation (Fig. 2n). Other compensating mutations in Omicron variants likely rescued the negative effects of S371L on ZCB11.

The potency and breadth of ZCB11 is determined by its unique structure to overcome mutations and deletions across current SARS-CoV-2 VOCs. We compared the interface regions between ZCB11 and similar RBD-specific class I NAbs derived from IGHV1-58 family, including B1-182.1[35] and S2E12[34,52]. ZCB11 uses a similar binding pattern as B1-182.1 and S2E12. Different from B1-182.1 and S2E12 engaging multiple binding sites with RBM, however, the CDR H3 of ZCB11 is the major region for interaction with RBM in the aforementioned first interface. The CDR H3 domains are highly conserved between S2E12 and B1-182.1 except for the single mutation of F108 in B1-182.1 replaced by S108 in S2E12. This S108 residue is responsible for higher neutralization potency against Omicron variants by avoiding potential steric repulsion upon binding to RBM[27]. Interestingly, ZCB11 has the same S108 as S2E12. Furthermore, sequence analysis of CDR H3 reveals several distinct residues in ZCB11 compared to S2E12 and B1-182.1. Structural analysis indicates that the unique RTI motif at positions 104-106 of ZCB11 is essential for stronger interaction with RBM. The salt bridge formed between D420 of RBM and R104 of ZCB11 shifts the

CDH R3 slightly to RBM, which does not exist in either S2E12 or B1-182.1. This arrangement enables the formation of the hydrogen bond between A475 of RBM and the main chain of C107 of ZCB11, further stabilizing the interaction between RBM and ZCB11. In addition, instead of conferring drug resistance, the S477N and T478K mutations in Omicron likely contributed to potency of ZCB11 by avoiding the steric clash found in some class I antibodies[27]. Meantime, L455, F456, and F486 in the RBM stabilized the second interface with ZCB11 by forming hydrophobic interactions. Taken together, these unique structural features likely contributed to higher potency of ZCB11 against pandemic Omicron variants. During manuscript revision, we have noticed that the newly emerged Omicron sublineages BA.4 and BA.5 in South Africa contain an F486V mutation, currently accounting for less than 0.01% of total sequences (https://www.gisaid.org/). The V486 residue retains the hydrophobic property in the second interface. Moreover, the first interface, relatively stronger one, is unlikely influenced by V486. Future experiments are needed to determine if V486 would cause a major conformational change in RBM, leading to the loss of ZCB11 broadly neutralizing activity.

ZCB11 represents one of the broadly reactive class I bNAbs reported thus far with comparable potency against all current SARS-CoV-2 VOCs including three Omicron variants tested. Most public antibodies were RBD-specific class I NAbs[21,32,33,36,53]. Accordingly, public antibody is encoded by B cell clonotypes isolated from different individuals that share similar genetic features[37]. In a previous study, 7 of 13 NAbs were found using IGHV3-53/3-66 heavy chain and paired predominantly with IGKV1-9*01 light chain[12]. These NAbs displayed abolished neutralizing activity after K417N in Beta variant was introduced into the pseudovirus neutralization assay. Interestingly, our newly cloned NAbs ZCB3, ZCC10, and ZCD3 use the same IGHV3-53/3-66 heavy chain but paired with IGKV1-9, IGKV3-20, and IGKV1-27 light chains, respectively. ZCB3, our second best bNAb, uses the identical pair of IGHV3-53/3-66 and IGKV1-9 but did not display neutralization reduction against the K417N pseudovirus. ZCB3, however, showed reduced neutralization potency for over 10-fold against Beta and Omicron variants as compared with ZCB11. Interestingly, ZCB11 uses IGHV1-58 heavy chain and IGKV3-20 light chain, which also belongs to public antibodies reported by other groups[32,34,35,37]. In these studies, patient-derived S2E12 and vaccine-induced 2C08 NAbs that shared 95% amino acid identity used the same IGHV1-58 heavy chain and IGKV3-20 light chain. 2C08 was able to prevent challenges against Beta and Delta variants in the hamster model. Like 82.2% amino acid identity between ZCB11 and S2E12, ZCB11 and 2C08 shared 83.8% amino acid identity in their heavy chain variable regions. The CDR H3 of 2C08 is

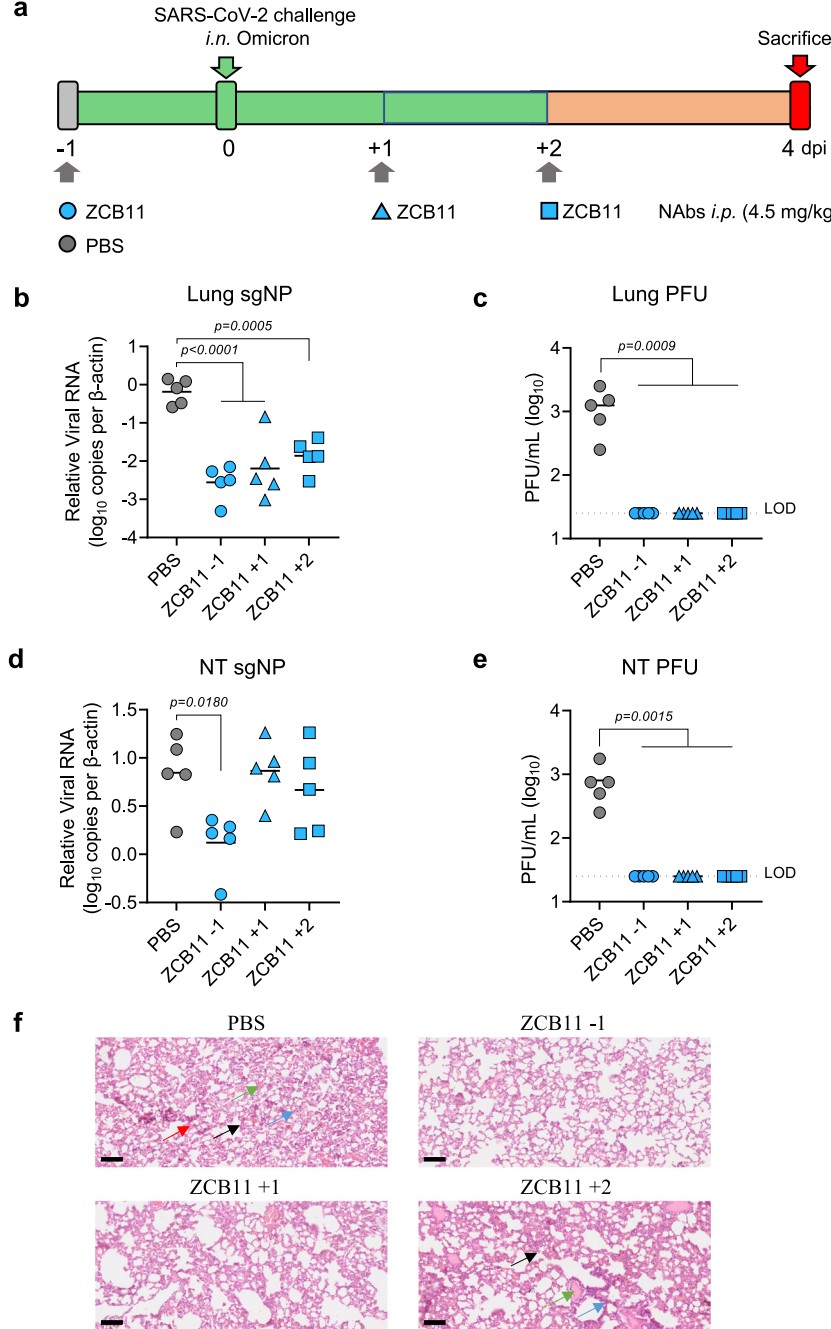

**Fig. 6 Therapeutic efficacy of ZCB11 against authentic SARS-CoV-2 Omicron BA.1 in golden Syrian hamsters. a** Experimental schedule and color coding for different treatment groups. Two groups of female hamsters ($n = 5$) received a single intraperitoneal injection of PBS (gray) or 4.5 mg/kg of ZCB11 (blue circle) at one day before viral infection (−1 dpi). 24 h later (day 0), all four groups of female hamsters were intranasally challenged with $10^5$ PFU live SARS-CoV-2 Omicron BA.1. The other two groups of female hamsters ($n = 5$) received a single intraperitoneal treatment of ZCB11 at 1 dpi (blue triangle) or 2 dpi (blue square), respectively. All animals were sacrificed on day 4 for final analysis. **b, d** The NP subgenomic RNA copy numbers (normalized by β-actin) in lung and NT homogenates of each group ($n = 5$) were determined by a sensitive RT-PCR. The detection of viral load was performed in at least triplicates. **c, e** Live viral plaque assay was used to quantify the number of infectious viruses in lung and NT homogenates. Log10-transformed PFU per mL were shown for each group ($n = 5$). The dash line indicates the limit of detection. **f** Representative histopathology of the lung tissues from treated female hamsters ($n = 5$) after viral challenge. Tissue sections were stained with hematoxylin and eosin (H&E). In the PBS-treated group and late ZCB11 treatment group (2 dpi), lung damages showed alveolar septa thickening (black arrow), some inflammatory cell infiltration (blue arrow), homogeneously pink foci of edema (green arrow) and multifocal hemorrhage (red arrow). The scale bar represents 100 μm. PFU plaque-forming unit, SEM standard error of the mean, NP nucleocapsid protein, NT nasal turbinate, LOD limit of detection. Each symbol represents an individual hamster with a line indicating the mean of each group. The color coding was consistently used in each graph. Statistics were generated using one-way ANOVA followed by Tukey's multiple comparisons test. *P*-values were shown on each graph where necessary. Source data (**b**-**e**) are provided as a Source Data file.

identical to S2E12 except for an P100A mutation. Although relatively high sequence homology was found in public B cell clonotypes encoded by IGHV1-58 among different human populations in the world, it remains unknow if vaccine design can elicit mainly high amounts of ZCB11-like bNAbs for protection. Notably, while BNT162b2-26 developed naturally the public bNAb ZCB11, it was unlikely dominantly elicited due to the reduced plasma neutralizing titer against Omicron BA.1 as compared with WT. To this end, future vaccine design should consider stabilizing the binding interface of the immunogen for interaction with ZCB11. Since ZCB11 protected hamsters against both the Delta and Omicron variants, the most dominant circulating SARS-CoV-2 VOCs in the world, our findings may warrant the clinical development of ZCB11 for immunotherapy and transmission prevention.

There are several limitations in this study. To understand the frequency of ZCB11-like bNAb among BNT162b2-vaccinees, we need to investigate other responders who show equally potent bNAb responses. We only tested single ZCB11 for prophylactic and therapeutic efficacy in the hamster model. Since resistant SARS-CoV-2 may readily emerge during antibody monotherapy[54], ZCB11 should be tested in cocktails by combining with other potent but non-competing bNAb (e.g., LY-CoV1404). For animal challenge experiments, the number of hamsters in each group was small mainly due to limited space capacity in our animal BSL-3 facility. We, however, have demonstrated the consistency of our SARS-CoV-2/hamster models in several studies[38,39]. Consistent results were also obtained for the repeated pre-treatment experiments in this study. Thus far, we have done a single-dose efficacy experiment. Since the number of infectious viruses in the PBS group of Delta-challenged hamsters was over one order of magnitude higher than that in the PBS group of Omicron-challenged animals, higher amount of ZCB11 might be needed to suppress subgenomic viral loads against the Delta variant. Different doses and routes of administration or antibody combination, therefore, should be tested in future experiments to provide useful information in support of clinical development of ZCB11 and ZCB11-like bNAb.

## Methods

**Ethics**. This acquisition of blood samples from vaccinated donors for identification of broad neutralizing activities and isolation of potent monoclonal antibodies against COVID-19 received approval from the Institutional Review Board of The University of Hong Kong/Hospital Authority Hong Kong West Cluster (Ref No. UW 21-120 452). The research was conducted in strict accordance with the rules and regulations of the Hong Kong government for the protection of human subjects. The study subjects agreed and signed the written informed consents for research use of their blood samples and indirect identifiers.

**Human subjects**. A cohort of 34 vaccinees who received two doses of BNT162b2 before June 2021 were recruited for voluntary blood donations (Supplementary Table 1). The exclusion criteria include individuals with (1) documented SARS-CoV-2 infection, (2) high-risk infection history within 14 days before vaccination, (3) COVID-19 symptoms such as sore throat, fever, cough and shortness of breath. Clinical and laboratory findings were entered into a predesigned database. Written informed consent was obtained from all study subjects. Blood samples were collected by professional clinical doctors and separated into plasma and peripheral blood mononuclear cells (PBMCs) by Ficoll-Hypaque gradient centrifugation. All plasma samples were heat-inactivated at 56 °C for at least 30 min before 1st test.

**Live viruses**. Authentic SARS-CoV-2 included D614G (MT835143), Alpha (MW856794), Beta (GISAID: EPI_ISL_2423556), Gamma P.3 (GISAID: EPI_ISL_2423558) and Delta (hCoV-19/Hong Kong/HKU-210804-001/2021; GISAID: EPI_ISL_3221329). Omicron BA.1 hCoV-19/Hong Kong/HKU-691/2021 (HKU691) (GISAID accession number EPI_ISL_7138045), Omicron BA.1.1 hCoV-19/Hong Kong/HKU-344/2021 (HKU344-R346K) (GISAID accession number EPI_ISL_7357684) and Omicron BA.2 (GISAID: EPI_ISL_9845731) were isolated from the combined nasopharyngeal-throat swabs of 2 people with COVID-19 in Hong Kong, respectively[25]. All experiments involving live SARS-CoV-2 followed the approved standard operating procedures in the Biosafety Level 3 (BSL-3) facility at The University of Hong Kong[55,56].

**Cell lines**. HEK293T cells, HEK293T-hACE2 cells, and Vero-E6-TMPRSS2 cells were maintained in DMEM containing 10% FBS, 2 mM L-glutamine, 100 U/mL/mL penicillin and incubated at 37 °C in a 5% $CO_2$ setting[57]. Expi293F™ cells were cultured in Expi293™ Expression Medium (Thermo Fisher Scientific) at 37 °C in an incubator with 80% relative humidity and a 5% $CO_2$ setting on an orbital shaker platform at $125 \pm 5$ rpm/min (New Brunswick innova™ 2100) according to the manufacturer's instructions.

**ELISA analysis of plasma and antibody binding to RBD and trimeric spike**. The recombinant RBD and trimeric spike proteins derived from SARS-CoV-2 (Sino Biological) were diluted to final concentrations of 1 μg/mL, then coated onto 96-well plates (Corning 3690) and incubated at 4 °C overnight. Plates were washed with PBST (PBS containing 0.05% Tween-20) and blocked with blocking buffer (PBS containing 5% skim milk or 1% BSA) at 37 °C for 1 h. Serially diluted plasma samples or isolated monoclonal antibodies were added to the plates and incubated at 37 °C for 1 h. Wells were then incubated with a secondary goat anti-human IgG labeled with horseradish peroxidase (HRP) (1:5000 Invitrogen) TMB substrate (SIGMA). Optical density (OD) at 450 nm was measured by SkanIt RE6.1 with VARIOSKAN Lux (Thermo Scientific). Serially diluted plasma from healthy individuals or previously published monoclonal antibodies against SARS-CoV-2 (ZB8) were used as negative controls.

**Isolation of SARS-CoV-2 spike-specific IgG+ single memory B cells by FACS**. RBD-specific single B cells were sorted as previously described[58]. In brief, PBMCs from vaccinated individuals were collected and incubated with an antibody cocktail and a His-tagged spike (Sino Biological) protein for identification of spike-specific B cells. The cocktail consisted of the Zombie viability dye (Biolegend), CD19-Percp-Cy5.5, CD3-Pacific Blue, CD14-Pacific Blue, CD56-Pacific Blue, IgM-Pacific Blue, IgD-Pacific Blue, IgG-PE, CD27-PE-Cy7 (1ul/test BD Biosciences or Biolegend) and the recombinant SARS-CoV-2 spike-His described above. Two consecutive staining steps were conducted: the first one used an antibody and spike cocktail incubation of 30 min at 4 °C; the second staining involved staining with anti-His-APC (5ul/test Abcam) and anti-His-FITC antibodies (1ul/test Abcam) at 4 °C for 30 min to detect the His tag of the spike. The stained cells were washed and resuspended in PBS containing 2% FBS before being strained through a 70-μm cell mesh filter (BD Biosciences). SARS-CoV-2 spike-specific single B cells were gated as CD19 + CD27 + CD3-CD14-CD56-IgM-IgD-IgG+spike+ and sorted by FACSAria III cell sorter (BD) into 96-well PCR plates containing 10 μL of RNAase-inhibiting RT-PCR catch buffer (1 M Tris-HCl pH 8.0, RNase inhibitor, DEPC-treated water, Thermo Scientific). Plates were then snap-frozen on dry ice and stored at −80 °C until the reverse transcription reaction. The population analysis of antigen-specific memory B cells was performed by FlowJo V10.

**Single B cell RT-PCR and antibody cloning**. Single memory B cells isolated from PBMCs of vaccinated donors were cloned as previously described[59]. Briefly, one-step RT-PCR was performed on sorted single memory B cell with a gene-specific primer mix, followed by nested PCR amplifications and sequencing using the heavy chain and light chain-specific primers. Cloning PCR was then performed using heavy chain and light chain-specific primers containing specific restriction enzyme cutting sites (heavy chain, 5′-AgeI/3′-SalI; kappa chain, 5′-AgeI/3′-BsiWI). The PCR products were purified and cloned into the backbone of antibody expression vectors containing the constant regions of human Igγ1. The constructed plasmids containing paired heavy and light chain expression cassettes were co-transfected into 293 T cells (ATCC) grown in 6-well plates. Antigen-specific ELISA and pseudovirus-based neutralization assays were used to analyze the binding capacity to SARS-CoV-2 spike and the neutralization capacity of transfected culture supernatants, respectively.

**Genetic analysis of the BCR repertoire**. Heavy chain and light chain germline assignment, framework region annotation, determination of somatic hypermutation (SHM) levels (in nucleotides) and CDR loop lengths (in amino acids) were performed with the aid of the NCBI IgBlast tool (https://www.ncbi.nlm.nih.gov/igblast/). Sequences were aligned using Clustal W in the BioEdit sequence analysis package (V7.2). Antibody clonotypes were defined as a set of sequences that share genetic V and J regions as well as an identical CDR3.

**Antibody production and purification**. The paired antibody VH/VL chains were cloned into Igγ and Igk expression vectors using T4 ligase (NEB). Antibodies produced from cell culture supernatants were purified immediately by affinity chromatography using recombinant Protein G-Agarose (Thermo Scientific) according to the manufacturer's instructions to purify IgG. The purified antibodies were concentrated by an Amicon ultracentrifuge filter device (molecular weight cut-off 10 kDa; Millipore) to a volume of 0.2 mL in PBS (Life Technologies), and then stored at 4 °C or −80 °C for further characterization.

**Pseudovirus-based neutralization assay**. The neutralizing activity of NAbs was determined using a pseudotype-based neutralization assay as we previously described[31,39]. Briefly, The pseudovirus was generated by co-transfection of HEK

293 T cells with pVax-1-S-COVID19 and pNL4-3Luc_Env_Vpr, carrying the optimized spike (S) gene (QHR63250) and a human immunodeficiency virus type 1 backbone, respectively[31,39]. Viral supernatant was collected at 48 h post-transfection and frozen at −80 °C until use. The serially diluted monoclonal antibodies or sera were incubated with 200 TCID50 of pseudovirus at 37 °C for 1 h. The antibody-virus mixtures were subsequently added to pre-seeded HEK 293T-ACE2 cells. 48 h later, infected cells were lysed to measure luciferase activity using a commercial kit (Promega, Madison, WI). Half-maximal ($IC_{50}$) or 90% ($IC_{90}$) inhibitory concentrations of the evaluated antibody were determined by inhibitor vs. normalized response—4 Variable slope using GraphPad Prism 8 or later (GraphPad Software Inc.).

**Neutralization activity of monoclonal antibodies against authentic SARS-CoV-2.** The SARS-CoV-2 focus reduction neutralization test (FRNT) was performed in a certified Biosafety level 3 laboratory. Neutralization assays against live SARS-CoV-2 were conducted using a clinical isolate previously obtained from a nasopharyngeal swab from an infected patient[60]. The tested antibodies were serially diluted, mixed with 50 μL of SARS-CoV-2 ($1 \times 10^3$ focus forming unit/mL, FFU/mL) in 96-well plates, and incubated for 1 hour at 37 °C. Mixtures were then transferred to 96-well plates pre-seeded with $1 \times 10^4$/well Vero-E6 cells and incubated at 37 °C for 24 h. The culture medium was then removed, and the plates were air-dried in a biosafety cabinet (BSC) for 20 min. Cells were then fixed with a 4% paraformaldehyde solution for 30 min and air-dried in the BSC again. Cells were further permeabilized with 0.2% Triton X-100 and incubated with cross-reactive rabbit sera anti-SARS-CoV-2-N (1:5000) for 1 h at RT before adding an Alexa Fluor 488 goat anti-rabbit IgG (H + L) cross-adsorbed secondary antibody (1:1000 Life Technologies). The fluorescence density of SARS-CoV-2 infected cells were scanned using a Sapphire Biomolecular Imager (Azure Biosystems) and the neutralization effects were then quantified using Fiji software (NIH).

**Antibody binding kinetics and competition between antibodies measured by Surface Plasmon Resonance (SPR).** The binding kinetics and affinity of recombinant monoclonal antibodies for the SARS-CoV-2 RBD protein (Sino Biological) were analyzed by SPR (Biacore T200, Cytiva). Specifically, the SARS-CoV-2 RBD protein was covalently immobilized to a CM5 sensor chip via amine groups in 10 mM sodium acetate buffer (pH 5.0) for a final RU around 250. SPR assays were run at a flow rate of 10 uL/min in HEPES buffer. For conventional kinetic/dose-response, serial dilutions of monoclonal antibodies were injected across the RBD protein surface for 180 s, followed by a 900 s dissociation phase using a multi-cycle method. Remaining analytes were removed in the surface regeneration step with the injection of 10 mM glycine-HCl (pH 1.5) for 60 s at a flow rate of 30 μl/min. Kinetic analysis of each reference subtracted injection series was performed using the Biacore Insight Evaluation Software (Cytiva). All sensorgram series were fit to a 1:1 (Langmuir) binding model of interaction. Before evaluating the competition between antibodies, both the saturating binding concentrations of antibodies for the immobilized SARS-CoV-2 RBD protein were determined separately. In the competitive assay, antibodies at the saturating concentration were injected onto the chip with immobilized RBD protein for 120 s until binding steady state was reached. The other antibody also used at the saturating concentration was then injected for 120 s, followed by another 120 s of injection of antibody to ensure a saturation of the binding reaction against the immobilized RBD protein. The differences in response units between antibody injection alone and prior antibody incubation reflect the antibodies' competitive ability by binding to the RBD protein.

**Western blot.** Antibodies went through SDS-PAGE at the constant voltage of 120 V and then transferring at the constant current of 0.15 A. The transferred nitrocellulose membrane was removed from the transfer gel holder cassette and washed twice with distilled water for 5 min. The membrane was blocked with blocking buffer at room temperature for 30 min with continuous shaking. The membrane was subsequently washed four times with PBST for 15 min each time and then incubated with SARS-CoV-2 S1 or S2 with His tag (1:1000 Sino Biological) diluted in 5% BSA at 4 °C overnight with continuous shaking. The membrane was washed four times with continuous shaking in PBST for 15 min followed by incubation with fluorescence-linked secondary antibody (1:1000 DyLight 650 rabbit anti-His, Abcam) in blocking buffer at room temperature for 1 h with continuous shaking. After washing and drying, the blotted membrane was scanned by Sapphire Biomolecular Imager with Sapphire Capture Software V1.7.0319.0. (Thermo Scientific biosystems).

**Cryo-EM sample preparation.** The SARS-CoV-2 Omicron BA.1 variant spike ECD trimer proteins were purchased from Sino Biological Inc. (Cat: 40589-V08H26). For the formation of Spike-Fab complex, the Omicron BA.1 Spike trimer (2.2 mg/ml) was mixed with Fab (2.9 mg/ml) in 1x PBS buffer at a molar ratio of 1:1.2 (Spike monomer:Fab) and incubated on ice for 40 min. 3.5 μl sample of Spike-Fab complex was applied to glow-discharged holey carbon grids (300 mesh C Flat Au R1.2/1.3), and plunge frozen into liquid ethane using MKIV Vitrobot (Thermo Fisher Scientific) after blotting extra samples with filter paper for 4 s at 4 °C and 100% humidity.

**Cryo-EM data acquisition and processing.** For the Spike-Fab complex, the grids were loaded onto a Titan Krios G3i transmission electron microscope (Thermo Fisher Scientific) operated at 300 kV. Movies were recorded on a Gatan K3 Summit direct electron detector and a Bio Quantum energy filter with a 20 eV slit width. Movies were collected with a 4.5 s exposure, resulted a total dose of ~50 e-/Å$^2$ over 40 frames, in counting mode at a nominal magnification of 81,000x using a defocus range of −1.0 to −2.5 μm. EPU software v 2.10.0 was used for data collection. The 4,857 cryo-EM movies were patch motion corrected using MortionCor2[61]. The CTF parameters estimation, particle picking and 2D classification were conducted in cryoSPARC (v 2.15.0)[62]. Particles were further heterogeneously refined in cryoSPARC to separate two distinct states of the Spike-Fab trimer complex. State1 (3 u) displays three RBDs in 'up' conformation with Fab bound, and State2 (2u1d) displays two RBDs in 'up' conformation with Fab bound and one RBD in 'down' conformation with no Fab bound. Particles from State1 and State2 were then performed 3D classification ($k = 5$) without alignment in RELION 3.1[63], respectively. Particles of each 3D class from previous step were imported into cryoSPARC to conduct non-uniform refinement, which aligned the RBD-Fab region with the strongest density from each particle to the same position. A tight mask was created to cover the region of density map except RBD and Fab, and particle subtraction was applied to selected high-quality classes. Final map was iterated after local refinement to RBD-Fab region using previously aligned RBD-Fab region of all high-quality particles from two states.

**Cryo-EM structure modeling and refinement.** SARS-CoV-2 Omicron 1-RBD up Spike trimer (PDB code: 7TEI [https://www.rcsb.org/structure/7TEI]) and Fab homology structure built by SWISS-Model server[64] were used as initial models to dock into cryo-EM density map. The initial models were then adjusted manually using Coot v 0.9.5[65] and refined in Coot and Phenix v 1.20[66]. Structural analysis was conducted in UCSF Chimera v 1.15[67] and UCSF ChimeraX v 1.3[68].

**Biosafety.** All experiments involving live SARS-CoV-2 including viral challenge experiments in hamsters followed strictly the approved standard operating procedures (SOPs) of the Biosafety Level-3 facility at the Department of Microbiology, HKU. For viral inactivation, plasma samples from infected animals were heat-inactivated at 56 °C for 30 min, the tissue homogenates were treated with cell lysis buffer for viral load detection and the tissues were fixed in neutralized formalin for 24 h before sending out for subsequent histopathology analysis. For animal challenge experiments, approval was obtained from the HKU Committee on the Use of Live Animals in Teaching and Research (CULATR 5359-20 and 5518-20).

**Hamster model and experiments.** In vivo evaluation of monoclonal antibody ZB8 or ZCB11 in the established golden Syrian hamster model of SARS-CoV-2 infection was performed as described previously[38]. 6–10-week-old male and female hamsters were purchased from the Chinese University of Hong Kong Laboratory Animal Service Centre through the HKU Centre for Comparative Medicine Research. The animals were housed as 4–5 hamsters per cage with sawdust bedding and environment enrichment. The hamsters were housed with free access to standard pellet feed and water ad libitum until live virus challenge in our BSL-3 animal facility. The hamsters were randomized from different litters into experimental groups. Experiments were performed in compliance with the relevant ethical regulations[38]. For prophylaxis studies, 24 h before live virus challenge, three groups of hamsters were intraperitoneally administered with one dose of test antibody in phosphate-buffered saline (PBS) at the indicated dose. At day 0, each hamster was intranasally inoculated with a challenge dose of 100 μL of Dulbecco's Modified Eagle Medium containing $10^5$ PFU of SARS-CoV-2 Delta variant or Omicron BA.1 variant under anesthesia with intraperitoneal ketamine (200 mg/kg) and xylazine (10 mg/kg). For therapeutics studies, the antibodies were intraperitoneally injected at day 1 or 2 post virus challenge. The hamsters were monitored daily for clinical signs of disease. Syrian hamsters typically clear virus within one week after SARS-CoV-2 infection. Accordingly, animals were sacrificed for analysis at day 4 after virus challenge with high viral loads[38]. Half the nasal turbinate, trachea, and lung tissues were used for viral load determination by quantitative RT-qPCR assay using QuanStudio Real-Time PCR system with the primers (5′-cgca-tacagtcttrcaggct-3′ and 5′-gtgtgatgttgawatgacatggtc-3′ for RdRp and 5′-cgatctcttg-tagatctgttctc-3′ and 5′-tccttgccatgttgagtgag-3′ for sgNP)[69]. Quantification of infectious virus was performed by plaque assay as we described previously[38]. Briefly, serial 10-fold dilutions of each tissue homogenate were inoculated in a Vero-E6 cell monolayer in quadruplicate and cultured in DMEM supplemented with 1% FBS and penicillin/streptomcycin. The plates were observed for cytopathic effects at day 4. Plaque-forming units (PFUs) were calculated by the number of plaques multiplied by the dilution factor and expressed as PFU/ml of tissue homogenate.

**Histopathological analysis and immunofluorescence (IF) staining.** Lung sections were collected, fixed in 4% formaldehyde solution and embedded in paraffin for the H&E and immunofluorescence staining. The whole tissue sections after H&E staining were scanned and analyzed using the Akoya Vectra Polaris™ Automated Quantitative Pathology Imaging System. The IF staining was conducted for identification and localization of SARS-CoV-2 nucleocapsid protein (NP) using a

rabbit anti-SARS-CoV-2-N antibody (1:5000) together with Alexa Fluor 488-conjugated goat anti-rabbit IgG antibody (1:1000 Life Technologies) as we previously described[39]. The images of the lung tissue section were captured using the Carl Zeiss LSM 900 confocal microscope and analyzed using the ZEN 3.3 software (Blue edition).

**Quantification and statistical analysis**. Statistical analysis was performed using PRISM 8.0 or later. Ordinary one-way ANOVA and multiple comparisons were used to compare group means and differences between multiple groups. Unpaired Student's $t$-tests were used to compare group means between two groups only. A $P$-value < 0.05 was considered significant. The number of independent replicates performed, the number of animals in each group, and the specific details of statistical tests are reported in the figure legends and the Methods section.

**Reporting summary**. Further information on research design is available in the Nature Research Reporting Summary linked to this article.

## Data availability

Data generated or analyzed during this study are included in this published article including supplementary information file. Source data are provided with this paper. We are applying for patent protection for some of the antibodies. All other information are also available from the corresponding author upon reasonable requests. Cryo-EM density maps of the spike-Fab complex have been deposited in the Electron Microscopy Data Bank (EMD-33195 for 3 u, EMD-33194 for 2u1d, respectively). Atomic coordinates have been deposited in the Protein Data Bank with the accession code 7XH8. Source data are provided with this paper.

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

## Acknowledgements

This study was supported by the Hong Kong Research Grants Council Collaborative Research Fund (C7156-20G, C1134-20G, and C5110-20G) and Shenzhen Science and Technology Program (JSGG20200225151410198 and JCYJ20210324131610027); the Health@InnoHK, Innovation and Technology Commission, The Government of the Hong Kong Special Administrative Region; and the National Program on Key Research Project of China (2020YFC0860600, 2020YFA0707500 and 2020YFA0707504); and donations from the Friends of Hope Education Fund in Hong Kong. Z.C.'s team was also partly supported by the Hong Kong Theme-Based Research Scheme (T11-706/18-N) and Wellcome Trust P86433. This study was also partly supported by funding the Health and Medical Research Fund, the Food and Health Bureau, The Government of the Hong Kong Special Administrative Region (Ref no.: COVID1903010-Project 7, COVID190123, 20190572 and 19181012); the Consultancy Service for Enhancing Laboratory Surveillance of Emerging Infectious Diseases and Research Capability on Antimicrobial Resistance for Department of Health of the Government of the Hong Kong Special Administrative Region; Sanming Project of Medicine in Shenzhen, China (grant no. SZSM201911014); the High Level-Hospital Program, Health Commission of Guangdong Province, China; the Major Science and Technology Program of Hainan Province (ZDKJ202003); and the research project of Hainan academician innovation platform (YSPTZX202004); and donations from the Shaw Foundation of Hong Kong, the Richard Yu and Carol Yu, Michael Seak-Kan Tong, May Tam Mak Mei Yin, Lee Wan Keung Charity Foundation Limited, the Providence Foundation Limited (in memory of the late Lui Hac Minh), Hong Kong Sanatorium & Hospital, Hui Ming, Hui Hoy and Chow Sin Lan Charity Fund Limited, Chan Yin Chuen Memorial Charitable Foundation, Marina Man-Wai Lee, the Hong Kong Hainan Commercial Association South China Microbiology Research Fund, the Jessie & George Ho Charitable Foundation, Perfect Shape Medical Limited, Kai Chong Tong, Tse Kam Ming Laurence, Foo Oi Foundation Limited, Betty Hing-Chu Lee, Ping Cham So, and Lo Ying Shek Chi Wai Foundation. All cryo-EM data were collected at the Biological Cryo-EM Center at the Hong Kong University of Science and Technology (HKUST), generously supported by a donation from the Lo Kwee Seong Foundation. S.D. acknowledges support from the Research Grants Council (RGC) of Hong Kong (ECS26101919, GRF16103321, C7009-20GF, C6001-21EF), Southern Marine Science and Engineering Guangdong Laboratory (Guangzhou) (SMSEGL20SC01-L), Guangdong Natural Science Foundation (GDST21SC04), Shenzhen Science and Technology Innovation Committee (SZ-SZSTI2108) and HKUST start-up and initiation grants. The funding sources had no role in the study design, data collection, analysis, interpretation, or writing of the report. We thank Drs. David D. Ho and Pengfei Wang for kindly providing the expression plasmids encoding for D614G, B.1.1.7, and B.1.351 variants and Dr. Linqi Zhang for B.1.617.2.

## Author contributions

Z.C. conceived and supervised this study. B.Z., R.Z. and Z.C. designed the experiments. B.Z., M.L. and B.C. cloned and characterized the NAbs. R.Z. did live viral neutralization and viral loads experiments. S.D. supervised the structural study. B.T., H.L., and S.D. solved the structure of ZCB11. J.F.-W.C., V.K.-M.P., C.C.-S.C., J.O.-L.T., C.C.-Y.C. designed and conducted hamster experiments. Q.P. performed pseudovirus neutralization and mutation mapping experiments. S.Y. and R.Z. did live viral plaque assay. R.Z., B.W.-Y.M., P.W., H.C., L.L., H.C. isolated and provided authentic viruses. M.L., B.C., H.-O.M., and K.-K.A. did plasmid cloning, ELISA, and RNA extraction. K.K.-W.T. provided clinical specimens. K.-Y.Y. supported the study with experimental resources.

## Competing interests

The authors declare no conflicts of interest except for a provisional patent application filed for human monoclonal antibodies generated in our laboratory with Z.C., B.Z., and R.Z. as the co-inventors. J.F.-W.C. has received travel grants from Pfizer Corporation Hong Kong and Astellas Pharma Hong Kong Corporation Limited. The funding sources had no role in study design, data collection, analysis or interpretation, or writing of the report.
