## [Peer Review File · Nature Communications]

Reviewer comments

Reviewer #1 (Remarks to the Author):

Review of Zhou et al

This concise manuscript aimed to address a limitation of many current therapeutic antibodies that are being clinically used against SARS-CoV-2, namely their loss of inhibitory activity against the Omicron variant due to the extensive group of mutations in the spike protein. The authors screened 34 donors who had received two doses of BNT162b2 mRNA vaccine for broadly neutralizing activity against a range of pseudoviruses with variant spike proteins. After identifying one optimal donor, they sorted spike-specific memory B cells from blood and produced a small number of mAbs that showed binding to spike protein. From this, they identified one primary candidate (ZCB11) that showed broad binding and neutralizing activity, the latter using both pseudovirus and live virus neutralization assays (IC50 values ~of 85.1, 39.9, 56.9, 11.2, 36.8 and 11.7 ng/mL for Alpha, Beta, Gamma, Delta, Omicron and Omicron R346K variants, respectively). Mapping studies showed limited escape with pseudoviruses containing S371L and a few RBM mutations (e.g., 493 and 505), although this loss of activity was not apparent in the context of the full Omicron spike, which had these substitutions. Based on sequencing and alignment results, they identify that ZCB11 belongs to a public clonotype (IgHV1-58 and IgKV3-20) and is related (~82% in the heavy chain) to a prior mAb (2E12) that has been analyzed structurally and shown to engage residues in the receptor-binding motif (RBM) of the receptor-binding domain (RBD); they infer that ZCB11 likely binds a similar or overlapping epitope. Finally, they performed prophylaxis studies in hamsters and showed that ZCB11 could confer protection against both Delta and Omicron variants. They conclude that ZCB11 is a promising elite and broadly neutralizing mAb for immunotherapy against pandemic SARS-CoV-2 variants of concern.

The strength of the paper includes the identification of a broadly neutralizing human mAb that has activity against many variants of concern, including Omicron. This finding is important given that several existing combination antibody therapies (e.g., Regeneron, Lilly, and Celltrion) lose almost all activity against Omicron. Another relative strength is that they show ZCB11 has activity in vivo in an animal model. Notwithstanding these points, there are some key weaknesses that dampen my enthusiasm, and require further supporting data. These include: (a) a lack of structural or additional functional data that corroborates their epitope assignment (which is based largely on relatedness to published 2E12 antibody); (b) a lack of direct functional comparison with 2E12, which could have been synthesized given its published sequence; (c) limited in vivo studies in hamsters that do not include post-exposure therapy, which would seem to be required to establish the potential of ZCB11 as an immunotherapy as suggested.

Major Comments

1. Mapping Results. The identification of a broadly inhibitory mAb against SARS-CoV-2 warrants a detailed determination of its epitope. How does it bind? Why does it still neutralize the current set of variants? Where is its liability for future variants? The current data set (Figure 3) is inadequate as it relies on a set of pseudoviruses with Omicron/Alpha/Beta/Gamma/Delta mutations that ultimately did not define a marked loss of activity (beyond the 11-fold reduction with S371L, the structural basis of which is not explained), a modeling analysis of a related S2E12 mAb with shared heavy and light chains, and competition binding studies. The authors acknowledge that greater definition of the epitope is needed (lines 274-276) – unlike the authors, this reviewer believes it is required in this current paper. Several of the following experiments should be pursued and included to enhance an understanding of the functional epitope of this broadly neutralizing mAb and what explains its unique profile of activity: (a) some type of landscape mutagenesis of spike and evaluation for loss-of binding [see (Greaney et al., 2021a; Greaney et al., 2021b)]; (b) neutralization escape studies with either authentic or chimeric SARS-CoV-2 viruses (Liu et al., 2021); and/or (c) crystallography or cryo-electron microscopy with ZCB11 and spike or RBD.

2. The in vivo data in Figure 4. The authors show protection against Delta and Omicron in hamsters with a very small $n = 4$ and establish activity only as prophylaxis. Several experiments are needed to buttress and expand this data: (a) For an antibody to have potential as an immunotherapy (which the authors repeatedly claim), the authors must show reasonable activity when administered post-exposure (e.g., at Day +1 or Day +2); (b) All animal experiments should be repeated independently and not reflect single experiments; (c) The authors should extend findings to the upper respiratory tract (nasal wash or turbinates) in addition to lung data; and (d) Finally, the data in Fig 4c and d seems to be at odds: why are subgenomic (replicating) RNA and infectious virus levels be so different (the sgRNA levels of 3 of 4 animals treated with ZCB11 do not look much different than the controls).

Other Comments.

1. In Figure 3, the authors make a comparison with 2E12 because of its shared heavy and light chains. Given the availability of the sequence, could they synthesize the mAb recombinantly and compare its potency and breadth head-to-head with ZCB11?

2. In many places, the authors term this antibody 'elite' neutralizing. However, with authentic virus, the IC50 values are 11 to 85 ng/ml against variants of concern. When this reviewer considers the definition of 'elite' neutralizing mAbs, the IC50 values should be below 10 ng/ml. Thus, this term should be removed from the Title, Abstract, Results, and Discussion.

3. Lines 55-56. Update infection and mortality numbers.

4. Lines 81, 85, 87, 94, 103, and elsewhere. What is an "elite vaccinee"? This term is not correct and should be replaced with "vaccine recipients having strongly neutralizing antibody responses".

5. Supplementary Figure 1a. Why were so few ($n = 14$) spike-specific mAbs recovered? In the FITC-spike vs APC-spike plots, (a) why are there not cells on the true diagonal? Is most of the staining background? There appears to be very little background in the naïve sample. Why is most of the staining single spike color positive?

6. Line 117 and Supplementary Figure 2c. The data (signal) establishing ZCB8 and ZCB9 as S2-specific mAbs is not altogether compelling. How much over the background was the signal? This should be defined. Did the mAbs work by Western blotting or some orthogonal assay?

7. Line 120. Delete "eventually".

8. Lines 122-124. It is difficult to conclude this given the issues with their spike protein and small numbers of mAbs recovered. This statement should be phrased differently or better, deleted entirely.

9. Lines 141-142. The authors cannot make these generalizations about BNT162b2 mRNA vaccine and public antibodies with just 4 mAbs. This statement should be deleted.

10. Line 150. "Most alarming" is not needed and should be deleted.

11. Lines 167-169. This statement about why ZCB11-like antibodies were not dominant in polyclonal responses is unclear and does not account for possible competition with less neutralizing antibodies. The statement should be edited for clarity and accuracy or deleted.

12. Line 173. The authors refer to classes I-IV or RBD mAbs here and elsewhere. They should define these in a structural image in a Figure or at a minimum define their characteristics in the text.

13. In Figure 3A, the authors define neutralization as fold change relative to D614G. Why are the loss of neutralization values (red) negative? Shouldn't they be positive – i.e., IC50 values of 10 and 5000 ng/ml for WT and E484K would be a 500-fold difference (5000/10).

14. Line 181-182. The sentence starting "Only and" does not make sense and needs editing.
15. Lines 215 and 221. "SASR" should be "SARS".
16. Lines 219-220 and Figure 4d. Given the limit of detection, the drop can only be four orders of magnitude not six.
17. Line 222-223. The statement on weaker pathogenicity of Omicron in hamsters should cite recently published papers (PMID: 35062015 and PMID: 35066015).
18. Line 242. "Till now" should "Until now".
19. Lines 244-246 and elsewhere. The authors state their identification of ZCB11 will have important implications on vaccine design. This statement is not substantiated or explained. How exactly do they imagine this would happen? It should be noted that the polyclonal sera from their vaccinated donor who made ZCB11 had relatively poorly neutralizing polyclonal antibodies against Omicron. So even if they make ZCB11 antibodies, will they function?
20. Line 257. The beginning of this sentence is a dangling clause and should be edited.
21. Lines 305-307. The authors should rephrase and avoid making such blanket claims about breadth given the existence of other published broadly neutralizing mAbs (e.g., S309 and PMID: 34261126).
22. Lines 494-495. Were unpaired t tests used in this paper?
23. In all Main and Supplemental Figure legends, the authors need to clearly define the number of independent experiments and the technical replicates within an experiment.
24. Figure 2b. The x-axis labels should be added to all graphs for consistency.

ADDITIONAL LITERATURE CITED.

Greaney, A.J., Starr, T.N., Barnes, C.O., Weisblum, Y., Schmidt, F., Caskey, M., Gaebler, C., Cho, A., Agudelo, M., Finkin, S., et al. (2021a). Mapping mutations to the SARS-CoV-2 RBD that escape binding by different classes of antibodies. *Nat Commun* 12, 4196.

Greaney, A.J., Starr, T.N., Gilchuk, P., Zost, S.J., Binshtein, E., Loes, A.N., Hilton, S.K., Huddleston, J., Eguia, R., Crawford, K.H.D., et al. (2021b). Complete Mapping of Mutations to the SARS-CoV-2 Spike Receptor-Binding Domain that Escape Antibody Recognition. *Cell Host Microbe* 29, 44-57.e49.

Liu, Z., VanBlargan, L.A., Bloyet, L.M., Rothlauf, P.W., Chen, R.E., Stumpf, S., Zhao, H., Errico, J.M., Theel, E.S., Liebeskind, M.J., et al. (2021). Identification of SARS-CoV-2 spike mutations that attenuate monoclonal and serum antibody neutralization. *Cell Host Microbe* 29, 477-488.e474.

Reviewer #2 (Remarks to the Author):

Zhou, B. et al. (2022) screened BNT162b2 vaccinated individuals for someone whose plasma could neutralize many variants of concern (VOC). A couple isolated antibodies from the identified individual could neutralize many VOC pseudotypes: ZCB3 and ZCB11. Only ZCB11 could still effectively neutralize the Omicron variant. Screening individual mutations in the spike protein revealed that S371L caused reduction in ZCB11 neutralization on its own, but not in the constellation of Omicron mutations. No tested mutations caused dramatic reductions in ZCB3

neutralization. ZCB11 protected hamsters from challenge with the Delta and Omicron variants.

Overall, the manuscript represents a large sum of work that yielded an antibody that could have therapeutic benefit. However, without further experiments, the novelty of this antibody is not clear. Specific comments are listed below.

MAJOR ISSUES:

Fig 3a: there is missing supplemental graphs showing the neutralization plots for all of these datapoints. In their absence, it is impossible to know the error associated with the values shown. For example, several NTD and S2 mutations cause drops in neutralization for RBD-specific antibodies; this is highly unexpected. I suspect that these drops are within the error of the assay performed (in which case, the -4 fold drop observed for ZCB3 with Q493R is not notable enough to mention in the text). Please show the missing neutralization plots.

Fig 3b: If ZCB11 and ZB3 binding sites overlap, and neither are impacted by the Omicron mutations, this site is an important target on the RBD. Please characterize where these antibodies are binding by either binding-competition experiments (similar to Fig 3b) with structurally characterized antibodies, or structure determination if possible. Sotrovimab, for instance, can also neutralize all VOC; competition with this NAb might demonstrate that they are targeting the same site.

Line 191-200: using a homology modeling program or percent identity to predict where an antibody is going to bind is unacceptable (unless the identity is 100%, which it is not).

OTHER ISSUES:

Line 95: do not use the word 'significantly' which implies that a statistical test was performed (unless a statistical test was performed)

Line 117: ZCB8 and ZCB9 are described as S2-specific, but the binding curves for these are not strong enough to support this.

Line 184: S371L caused a decrease in ZCB11 binding, but not in the constellation of Omicron mutations. This contradiction might be resolved if the authors model what a single S371L mutation might look like, compared with known structures of the Omicron RBD. Neighboring mutations S373P and S375F might change the orientation of S371L, for example. Also, please include the numerical reductions in IC50 for VOC with all mutations in Fig 3a; this would have been calculated in Fig. 2.

Line 235: Sotrovimab can potentially neutralize all current SARS-CoV-2 VOC, including the Omicron variant. Reword this sentence.

Line 253: N501Y does not confer resistance to NTD-specific Nabs.

Line 258: the Delta variant has more than just 4 spike mutations, and only 2 of the listed mutations here confer antibody evasion; so, it is not clear why these 4 mutations were chosen.

Line 298: Please explain how this clonotype was found in different ethnic human populations. What does this statement mean?

Line 290-295: Why are the percent identities shown? What is the implication the authors are trying to convey here?

Reviewer #3 (Remarks to the Author):

The manuscript entitled 'An elite broadly neutralizing antibody protects SARS-CoV-2 Omicron variant challenge' by Zhou and colleagues describes the development of a broadly neutralizing monoclonal antibody (bNAb; ZCB11). ZCB11 targets the viral receptor-binding domain and neutralizes all authentic SARS-CoV-2 VOCs including Omicron. The authors conclude that ZCB11 serves as a promising bNAb against pandemic SARS-CoV-2 VOCs.

Strengths: The strategy for selecting and characterizing broadly neutralizing monoclonal antibodies is well done and a strength of this manuscript.

Weaknesses: The animal work is weak and could have been designed more appropriately, especially regarding Omicron given the increased transmissibility of this VOC.

Major Concerns

- (1) The animal groups are small with only four hamsters. The authors should provide a power analysis for the animal group numbers.
- (2) The animal study should be improved to address the characteristics of the Omicron VOC. As increased transmissibility is one trade of Omicron, the authors should have collected upper respiratory tract samples such as oral swabs for subgenomic RNA and infectious titer determination. An additional sample type of the upper (nasal turbinate) or lower (trachea) respiratory tract would have been helpful to further address virus shedding and replication.
- (3) Histopathology of respiratory tract tissue (nasal turbinate, trachea, lung) would have been helpful here to confirm the effect of the bNAb ZCB11.
- (4) The authors should further elaborate how their findings affects future vaccine design (Discussion).
- (5) The authors should address the issue of treatment with a single monoclonal antibody. This is briefly touched und 'limitations' but likely should get its on paragraph in the Discussion.

Minor Concerns:

- (1) The Introduction needs update.
- (2) The Discussion needs a conclusions paragraph.
- (3) Methods should have its own biosafety paragraph addressing approval of biocontainment level and standard operating procedures for sample inactivation.
- (4) Methods should likely have a separate paragraph on animal study approval (currently one sentence under 'Hamster experiments') and animal welfare issues such as holding, food, water and environmental conditions should be addressed.

REVIEWER COMMENTS

Reviewer #1 (Remarks to the Author):

Review of Zhou et al

This concise manuscript aimed to address a limitation of many current therapeutic antibodies that are being clinically used against SARS-CoV-2, namely their loss of inhibitory activity against the Omicron variant due to the extensive group of mutations in the spike protein. The authors screened 34 donors who had received two doses of BNT162b2 mRNA vaccine for broadly neutralizing activity against a range of pseudoviruses with variant spike proteins. After identifying one optimal donor, they sorted spike-specific memory B cells from blood and produced a small number of mAbs that showed binding to spike protein. From this, they identified one primary candidate (ZCB11) that showed broad binding and neutralizing activity, the latter using both pseudovirus and live virus neutralization assays (IC50 values ~of 85.1, 39.9, 56.9, 11.2, 36.8 and 11.7 ng/mL for Alpha, Beta, Gamma, Delta, Omicron and Omicron R346K variants, respectively).

Mapping studies showed limited escape with pseudoviruses containing S371L and a few RBM mutations (e.g., 493 and 505), although this loss of activity was not apparent in the context of the full Omicron spike, which had these substitutions. Based on sequencing and alignment results, they identify that ZCB11 belongs to a public clonotype (IGHV1-58 and IgKV3-20) and is related (~82% in the heavy chain) to a prior mAb (2E12) that has been analyzed structurally and shown to engage residues in the receptor-binding motif (RBM) of the receptor-binding domain (RBD); they infer that ZCB11 likely binds a similar or overlapping epitope. Finally, they performed prophylaxis studies in hamsters and showed that ZCB11 could confer protection against both Delta and Omicron variants. They conclude that ZCB11 is a promising elite and broadly neutralizing mAb for immunotherapy against pandemic SARS-CoV-2 variants of concern.

The strength of the paper includes the identification of a broadly neutralizing human mAb that has activity against many variants of concern, including Omicron. This finding is important given that several existing combination antibody therapies (e.g., Regeneron, Lilly, and Celltrion) lose almost all activity against Omicron. Another relative strength is that they show ZCB11 has activity in vivo in an animal model. Notwithstanding these points, there are some key weaknesses that dampen my enthusiasm, and require further supporting data. These include: (a) a lack of structural or additional functional data that corroborates their epitope assignment (which is based largely on relatedness to published 2E12 antibody); (b) a lack of direct functional comparison with 2E12, which could have been synthesized given its published sequence; (c) limited in vivo studies in hamsters that do not include post-exposure therapy, which would seem to be required to establish the potential of ZCB11 as an immunotherapy as suggested.

Response: We thank the reviewer for his supportive comments and helpful suggestions. We have addressed these three key weaknesses carefully by providing (a) structural or additional functional data on ZCB11 (Fig. 4, Supplementary Fig. 6); (b) a direct neutralization comparison with S2E12 and three similar antibodies 2C08, B1-182.1 and COV2-2196 (revised Fig. 3c); (c) additional in vivo studies in hamsters for post-exposure therapy (Fig. 6).

Major Comments

1. *Mapping Results. The identification of a broadly inhibitory mAb against SARS-CoV-2 warrants a detailed determination of its epitope. How does it bind? Why does it still neutralize the current set of variants? Where is its liability for future variants? The current data set (Figure 3) is inadequate as it relies on a set of pseudoviruses with Omicron/Alpha/Beta/Gamma/Delta mutations that ultimately did not define a marked loss of activity (beyond the 11-fold reduction with S371L, the structural basis of which is not explained), a modeling analysis of a related S2E12 mAb with shared heavy and light chains, and competition binding studies. The authors acknowledge that greater definition of the epitope is needed (lines 274-276) – unlike the authors, this reviewer believes it is required in this current paper. Several of the following experiments should be pursued and included to enhance an understanding of the functional epitope of this broadly neutralizing mAb and what explains its unique profile of activity: (a) some type of landscape mutagenesis of spike and evaluation for loss-of binding [see (Greaney et al., 2021a; Greaney et al., 2021b)]; (b) neutralization escape studies with either authentic or chimeric SARS-CoV-2 viruses (Liu et al., 2021); and/or (c) crystallography or cryo-electron microscopy with ZCB11 and spike or RBD.*

Response: We have performed Cryo-EM analysis of ZCB11 with binding to the spike of SARS-CoV-2 Omicron variant and compared the binding epitope of ZCB11 with that of S2E12 (Fig. 4, Supplementary Fig. 6). We found that ZCB11 belongs to the class I antibody that binds to the tip of RBD like other VH1-58 antibodies. We then have performed head-to-head comparisons between ZCB11 and other potent VH1-58 antibodies including S2E12, 2C08, B1-182.1 and COV2-2196 on binding affinity (Fig. 3b). Both competition for binding to RBD (Supplementary Fig. 4e to 4h) and neutralization against Omicron variant were determined (Fig. 3c). ZCB11 showed strong competition with the other VH1-58 public antibodies by the SPR analysis, indicating their shared binding epitopes. Moreover, ZCB11 displayed a fast-on-slow-off binding mode to RBD, whereas other VH1-58 public antibodies disassociated faster after binding to RBD (Fig. 3b). In addition, we found that ZCB11 displayed much better neutralizing activity compared with other VH1-58 public antibodies against the Omicron variant (Fig. 3c).

2. *The in vivo data in Figure 4. The authors show protection against Delta and Omicron in hamsters with a very small n = 4 and establish activity only as prophylaxis. Several experiments are needed to buttress and expand this data: (a) For an antibody to have potential as an immunotherapy (which the authors repeatedly claim), the authors must show reasonable activity when administered post-exposure (e.g., at Day +1 or Day +2); (b) All animal experiments should be repeated independently and not reflect single experiments; (c) The authors should extend findings to the upper respiratory tract (nasal wash or turbinates) in addition to lung data; and (d) Finally, the data in Fig 4c and d seems to be at odds: why are subgenomic (replicating) RNA and infectious virus levels be so different (the sgRNA levels of 3 of 4 animals treated with ZCB11 do not look much different than the controls).*

Response: We fully agree with these helpful suggestions. During revision, we have repeated the prophylaxis experiment and performed post-exposure treatment experiments at 1 dpi and 2 dpi with the animal number of 5 (Fig. 6). We have also determined the infectious virus and viral loads in the nasal turbinate (NT) in addition

to the lung (Fig. 5e, 5f, 5j and 5k). As for the difference between sgRNA and PFU, sgRNA represents viral RNA released by replicating virus whereas PFU stands for released live infectious virus. Some research group suggested that sgRNA may not be an indicator for active live viral replication because it has nuclease resistance¹. For significantly reduced PFU, it is possible that neutralizing antibodies left in the tissue homogenate might inhibit live viral infection but have no effects on released tissue viral RNA.

Other Comments.

1. *In Figure 3, the authors make a comparison with S2E12 because of its shared heavy and light chains. Given the availability of the sequence, could they synthesize the mAb recombinantly and compare its potency and breadth head-to-head with ZCB11?*

Response: Thanks for the helpful suggestion. We have now made direct comparison between ZCB11 and S2E12 (revised Fig. 3c). As shown in this Figure, we have also provided direct parallel comparison with other VH1-58 public antibodies.

2. *In many places, the authors term this antibody ‘elite’ neutralizing. However, with authentic virus, the IC₅₀ values are 11 to 85 ng/ml against variants of concern. When this reviewer considers the definition of ‘elite’ neutralizing mAbs, the IC₅₀ values should be below 10 ng/ml. Thus, this term should be removed from the Title, Abstract, Results, and Discussion.*

Response: Many publications define ultrapotent or elite neutralizing antibody using pseudotyped viruses for IC₅₀ values below 10 ng/ml. As described in our manuscript, the IC₅₀ value of ZCB11 was 6 ng/ml against Omicron pseudovirus comparable to 5 ng/ml of LY-CoV1404 tested in parallel, belonging to one of the ‘elite’ neutralizing mAbs. During revision, we found that ZCB11 is more potent than other ultrapotent S2E12 or the B1-182.1 in neutralizing the Omicron variant (Fig. 3c), further supporting ZCB11 as one of the elite neutralizing antibodies.

3. *Lines 55-56. Update infection and mortality numbers.*

Response: Yes, have updated it in the manuscript (Lines 65-68 now).

4. *Lines 81, 85, 87, 94, 103, and elsewhere. What is an “elite vaccinee”? This term is not correct and should be replaced with “vaccine recipients having strongly neutralizing antibody responses”.*

Response: We agree and have removed the term of elite vaccinee.

5. *Supplementary Figure 1a. Why were so few (n = 14) spike-specific mAbs recovered? In the FITC-spike vs APC-spike plots, (a) why are there not cells on the true diagonal? Is most of the staining background? There appears to be very little background in the naïve sample. Why is most of the staining single spike color positive?*

Response: We received 5 million PBMCs from the donor to sort antigen-specific memory B cells and obtained a total of 40 spike-specific memory B cells. 14 mAbs were obtained from these 40 spike-specific memory B cells. The low number of double spike-specific memory B cells was probably related to donor, who had received the second vaccination for 130 days. For the staining data, most memory B cells with single spike-color positive was caused by the inappropriate compensation, and we have re-analysed the data as shown in Supplementary Fig. 1.

6. Line 117 and Supplementary Figure 2c. The data (signal) establishing ZCB8 and ZCB9 as S2-specific mAbs is not altogether compelling. How much over the background was the signal? This should be defined. Did the mAbs work by Western blotting or some orthogonal assay?

Response: The binding ability of ZCB8 and ZCB9 to single S2 subunit was very weak. By comparing their binding to different S subunits, we considered these two mAbs as S2-specific antibodies. We have performed Western blot to further test these two antibodies (Supplementary Fig. 2g). ZCB8 was indeed S2-specific, whereas ZCB9 was hardly detected, indicating the rather weak binding to S2 (Supplementary Fig. 2g). We have modified the text (Line 142 now) and supplementary tables (Supplementary Table 3) accordingly.

7. Line 120. Delete “eventually”.

Response: Agreed and deleted it.

8. Lines 122-124. It is difficult to conclude this given the issues with their spike protein and small numbers of mAbs recovered. This statement should be phrased differently or better, deleted entirely.

Response: Agreed and deleted it.

9. Lines 141-142. The authors cannot make these generalizations about BNT162b2 mRNA vaccine and public antibodies with just 4 mAbs. This statement should be deleted.

Response: Agreed and deleted it.

10. Line 150. “Most alarming” is not needed and should be deleted.

Response: Agreed and deleted it.

11. Lines 167-169. This statement about why ZCB11-like antibodies were not dominant in polyclonal responses is unclear and does not account for possible competition with less neutralizing antibodies. The statement should be edited for clarity and accuracy or deleted.

Response: Agreed and deleted it.

12. Line 173. The authors refer to classes I-IV or RBD mAbs here and elsewhere. They should define these in a structural image in a Figure or at a minimum define their characteristics in the text.

Response: Yes, we have added a paragraph in revised introduction to define these antibodies (Lines 92-97 now).

13. In Figure 3A, the authors define neutralization as fold change relative to D614G. Why are the loss of neutralization values (red) negative? Shouldn't they be positive – i.e., IC₅₀ values of 10 and 5000 ng/ml for WT and E484K would be a 500-fold difference (5000/10).

Response: We used the formula to calculate the fold change, (IC₅₀ of WT-IC₅₀ of mutant)/IC₅₀ of WT.

14. Line 181-182. The sentence starting “Only and” does not make sense and needs editing.

Response: Thank you for the suggestion. We have edited this sentence (Line 202 now).

15. Lines 215 and 221. “SASR” should be “SARS”.

Response: Sorry for the typo. We have corrected it.

16. Lines 219-220 and Figure 4d. Given the limit of detection, the drop can only be four orders of magnitude not six.

Response: Thanks for pointed out. We have corrected it in the text (Lines 275-277 now).

17. Line 222-223. The statement on weaker pathogenicity of Omicron in hamsters should cite recently published papers (PMID: 35062015 and PMID: 35066015).

Response: Agreed and cited these two papers (Line 282 now).

18. Line 242. “Till now” should “Until now”.

Response: Agreed and corrected it.

19. Lines 244-246 and elsewhere. The authors state their identification of ZCB11 will have important implications on vaccine design. This statement is not substantiated or explained. How exactly do they imagine this would happen? It should be noted that the polyclonal sera from their vaccinated donor who made ZCB11 had relatively poorly neutralizing polyclonal antibodies against Omicron. So even if they make ZCB11 antibodies, will they function?

Response: We have modified the sentences by saying that “future vaccine design should consider stabilizing the binding interface of the immunogen for interaction with ZCB11” (Lines 397-399 now).

20. Line 257. The beginning of this sentence is a dangling clause and should be edited.

Response: Agreed, we have corrected it (Line 333 now).

21. Lines 305-307. The authors should rephrase and avoid making such blanket claims about breadth given the existence of other published broadly neutralizing mAbs (e.g., S309 and PMID: 34261126).

Response: Agreed, we have removed this sentence.

22. Lines 494-495. Were unpaired t tests used in this paper?

Response: Yes, we used unpaired t tests to compare two groups.

23. In all Main and Supplemental Figure legends, the authors need to clearly define the number of independent experiments and the technical replicates within an experiment.

Response: We have added the information in the updated figure legends.

24. Figure 2b. The x-axis labels should be added to all graphs for consistency.

Response: Thanks, we have added the x-axis labels.

Reviewer #2 (Remarks to the Author):

Zhou, B. et al. (2022) screened BNT162b2 vaccinated individuals for someone whose plasma could neutralize many variants of concern (VOC). A couple isolated antibodies from the identified individual could neutralize many VOC pseudotypes: ZCB3 and ZCB11. Only ZCB11 could still effectively neutralize the Omicron variant. Screening individual mutations in the spike protein revealed that S371L caused reduction in ZCB11 neutralization on its own, but not in the constellation of Omicron mutations. No tested mutations caused dramatic reductions in ZCB3 neutralization. ZCB11 protected hamsters from challenge with the Delta and Omicron variants.

Overall, the manuscript represents a large sum of work that yielded an antibody that could have therapeutic benefit. However, without further experiments, the novelty of this antibody is not clear. Specific comments are listed below.

MAJOR ISSUES:

Fig 3a: there is missing supplemental graphs showing the neutralization plots for all of these datapoints. In their absence, it is impossible to know the error associated with the values shown. For example, several NTD and S2 mutations cause drops in neutralization for RBD-specific antibodies; this is highly unexpected. I suspect that these drops are within the error of the assay performed (in which case, the -4 fold drop observed for ZCB3 with Q493R is not notable enough to mention in the text). Please show the missing neutralization plots.

Response: We apologize for the missing graphs. We have now added the graphs in the supplementary Fig. 3.

Fig 3b: If ZCB11 and ZB3 binding sites overlap, and neither are impacted by the Omicron mutations, this site is an important target on the RBD. Please characterize where these antibodies are binding by either binding-competition experiments (similar to Fig 3b) with structurally characterized antibodies, or structure determination if possible. Sotrovimab, for instance, can also neutralize all VOC; competition with this NAb might demonstrate that they are targeting the same site. Line 191-200: using a homology modeling program or percent identity to predict where an antibody is going to bind is unacceptable (unless the identity is 100%, which it is not).

Response: We fully agreed and have obtained the Cryo-EM structure of ZCB11-Omicron spike complex. We now show that ZCB11 is also a class I antibody, which competes the binding epitopes with S2E12, 2C08, B1-182.1 and COV2-2196 in the same class as described in revised Fig. 3b and Fig. 4. With new structural figures added, we have removed the prediction figure generated by the homology modelling program.

OTHER ISSUES:

Line 95: do not use the word ‘significantly’ which implies that a statistical test was performed (unless a statistical test was performed)

Response: Agreed, we have revised it (Line 120 now).

Line 117: ZCB8 and ZCB9 are described as S2-specific, but the binding curves for these are not strong enough to support this.

Response: Agreed, we have used Western blot to further confirm their specificity (Supplementary Fig. 2g). ZCB8 was able to be blotted by SARS-CoV-2 S2, whereas ZCB9 was hardly blotted, further indicating rather weak binding to S2 (Supplementary Fig. 2g).

Line 184: S371L caused a decrease in ZCB11 binding, but not in the constellation of Omicron mutations. This contradiction might be resolved if the authors model what a single S371L mutation might look like, compared with known structures of the Omicron RBD. Neighbouring mutations S373P and S375F might change the orientation of S371L, for example. Also, please include the numerical reductions in IC50 for VOC with all mutations in Fig 3a; this would have been calculated in Fig. 2.

Response: We thank the reviewer for the helpful comments. Our Cryo-EM analysis results have now solved this puzzle. We found that ZCB11 binds to the up RBD domain, which was likely stabilized by the S371L and relevant mutations (Fig 4). We have added the numerical reductions in IC50 for VOCs with all mutations in Fig 3a.

Line 235: Sotrovimab can potently neutralize all current SARS-CoV-2 VOC, including the Omicron variant. Reword this sentence.

Response: Agreed and removed it.

Line 253: N501Y does not confer resistance to NTD-specific Nabs.

Response: Thanks, we have revised this sentence accordingly (Line 329 now).

Line 258: the Delta variant has more than just 4 spike mutations, and only 2 of the listed mutations here confer antibody evasion; so, it is not clear why these 4 mutations were chosen.

Response: Thank you for your suggestion. As shown in Fig. 3a, we tested all spike mutations of the Delta variant. We have revised this in Line 334 now.

Line 298: Please explain how this clonotype was found in different ethnic human populations. What does this statement mean?

Response: We mean that VH1-58 clonotype was found not just in Chinese vaccinee for generating potent bNAbs. We have removed this sentence on “ethnic” to avoid confusion (Lines 392-393 now).

Line 290-295: Why are the percent identities shown? What is the implication the authors are trying to convey here?

Response: We wanted to indicate that the sequence of ZCB11 is not 100% identical to other public VH1-58 neutralizing antibodies previously published.

Reviewer #3 (Remarks to the Author):

The manuscript entitled 'An elite broadly neutralizing antibody protects SARS-CoV-2 Omicron variant challenge' by Zhou and colleagues describes the development of a broadly neutralizing monoclonal antibody (bNAb; ZCB11). ZCB11 targets the viral receptor-binding domain and neutralizes all authentic SARS-CoV-2 VOCs including Omicron. The authors conclude that ZCB11 serves as a promising bNAb against pandemic SARS-CoV-2 VOCs.

Strengths: The strategy for selecting and characterizing broadly neutralizing monoclonal antibodies is well done and a strength of this manuscript.

Weaknesses: The animal work is weak and could have been designed more appropriately, especially regarding Omicron given the increased transmissibility of this VOC.

Major Concerns

(1) The animal groups are small with only four hamsters. The authors should provide a power analysis for the animal group numbers.

Response: We agree that the number of animals in each group is small, which is restricted by cage capacity of housing maximal 24 hamsters for each experiment in our BSL-3 animal facility. Therefore, we have been consistently using 4 hamsters per group throughout our hamster experiments as previously published^{2,3}. During revision, however, we have repeated the preventive experiment using additional 5 hamsters with consistent results obtained (Fig. 6).

(2) The animal study should be improved to address the characteristics of the Omicron VOC. As increased transmissibility is one trade of Omicron, the authors should have collected upper respiratory tract samples such as oral swabs for subgenomic RNA and infectious titer determination. An additional sample type of the upper (nasal turbinate) or lower (trachea) respiratory tract would have been helpful to further address virus shedding and replication.

Response: We fully agreed with the helpful suggestions. During revision, we have now examined specimens in the nasal turbinate. We have now included these results in Fig. 5 and Fig. 6.

(3) Histopathology of respiratory tract tissue (nasal turbinate, trachea, lung) would have been helpful here to confirm the effect of the bNAb ZCB11.

Response: Agreed. We have added the available histopathology data of lungs to further confirm the effects of ZCB11 in Fig. 5f and Fig. 6f.

(4) The authors should further elaborate how their findings affects future vaccine design (Discussion).

Response: Thanks for this suggestion. We have elaborated the implication of our findings for future vaccine design in discussion by saying that “future vaccine design should consider stabilizing the binding interface of the immunogen for interaction with ZCB11” (Lines 396-398 now).

(5) The authors should address the issue of treatment with a single monoclonal antibody. This is briefly touched und ‘limitations’ but likely should get its on paragraph in the Discussion.

Response: Thanks for this suggestion. We have addressed some issues of single mAb therapy in the discussion (Lines 404-407 now). We say that “Since resistant SARS-CoV-2 may readily emerge during antibody monotherapy⁴, ZCB11 should be tested in cocktails by combining with other potent but non-competing bNAb (e.g. LY-CoV1404).”

Minor Concerns:

(1) The Introduction needs update.

Response: We have updated the introduction part.

(2) The Discussion needs a conclusions paragraph.

Response: We have prepared the conclusion paragraph at the beginning of the discussion part (Lines 313-324 now).

(3) Methods should have its own biosafety paragraph addressing approval of biocontainment level and standard operating procedures for sample inactivation.

Response: Thanks for this suggestion. We have a separate paragraph addressing the biosafety issues and procedure of sample inactivation (Lines 612-620 now).

(4) Methods should likely have a separate paragraph on animal study approval (currently one sentence under ‘Hamster experiments’) and animal welfare issues such as holding, food, water and environmental conditions should be addressed.

Response: Thanks for this suggestion. We have added the animal welfare issues in the Methods (Lines 627-629 now).

- 1 Alexandersen, S., Chamings, A. & Bhatta, T. R. SARS-CoV-2 genomic and subgenomic RNAs in diagnostic samples are not an indicator of active replication. *Nature Communications* **11**, 6059, doi:10.1038/s41467-020-19883-7 (2020).
- 2 Zhou, D. *et al.* Robust SARS-CoV-2 infection in nasal turbinates after treatment with systemic neutralizing antibodies. *Cell Host Microbe* **29**, 551-563.e555, doi:10.1016/j.chom.2021.02.019 (2021).
- 3 Liu, L. *et al.* Potent neutralizing antibodies against multiple epitopes on SARS-CoV-2 spike. *Nature* **584**, 450-456, doi:10.1038/s41586-020-2571-7 (2020).
- 4 Ku, Z. *et al.* Molecular determinants and mechanism for antibody cocktail preventing SARS-CoV-2 escape. *Nature Communications* **12**, 469, doi:10.1038/s41467-020-20789-7 (2021).

Reviewer comments, second round review

Reviewer #1 (Remarks to the Author):

Review of Zhou et al

This revised manuscript addresses a limitation of many current therapeutic antibodies that are being clinically used against SARS-CoV-2, namely their loss of inhibitory activity against the Omicron variant due to the extensive group of mutations in the spike protein. The first version of the manuscript was generally strong, although some limitations were noted including an incomplete description of the ZCB11 binding epitope and rather limited protection studies in vivo in the hamster. The authors provided a detailed response in their rebuttal, and addressed virtually every concern raised by the Reviewers including the addition of cryo-EM data on the binding site of ZCB11 as well as new post-infection therapeutic studies that confirm its potential as an immunotherapy. The authors should be commended for their additional effort and excellent contribution to the field. I only have a few small comments.

Minor Comments.

1. The authors show that ZCB11 has interactions with residue F486 (as do other public clonotype antibodies). Although it is not a major contact, this residue could be a liability for ZCB11 against emerging BA.4 and BA.5 sublineage variants (F486V). Can the authors test ZCB11 against pseudoviruses with this mutation (ideally using the BA.4 or BA.5 sequence)? At a minimum, the authors should point out this residue as a possible one of concern in their limitations section in the context of loss of broadly neutralizing activity.

2. Histopathology. In Figures 5f and 6f, the images lack scale bars. Also, in the legends, the authors state representative images are used, but provide no indication as to how many animals were surveyed, which should be included. Finally, does Omicron really cause significant lung pathology in hamsters (e.g., see (Halfmann et al., 2022) as suggested in Figure 5? Why is there an apparent disparity in the lung disease severity of the PBS-treated and Omicron-infected hamsters in Figures 5 and 6. Were these harvested on the same day? There is no description of the histology in the Methods, and the level of detail in text main text or legends describing what findings are different in Figure 6f (ZCB11 versus PBS treatment) is lacking.

LITERATURE CITED.

Halfmann, P.J., Iida, S., Iwatsuki-Horimoto, K., Maemura, T., Kiso, M., Scheaffer, S.M., Darling, T.L., Joshi, A., Loeber, S., Singh, G., et al. (2022). SARS-CoV-2 Omicron virus causes attenuated disease in mice and hamsters. *Nature*.

Reviewer #2 (Remarks to the Author):

All of my concerns were addressed. The manuscript looks excellent.

Reviewer #3 (Remarks to the Author):

The authors responded appropriately to all my comments on the original manuscript. They performed additional experiments and added a second tissue target.

In regards to histopathology, additional immunohistochemistry targeting SARS-CoV-2 antigen would have been helpful.

REVIEWER COMMENTS

Reviewer #1 (Remarks to the Author):

Review of Zhou et al

This revised manuscript addresses a limitation of many current therapeutic antibodies that are being clinically used against SARS-CoV-2, namely their loss of inhibitory activity against the Omicron variant due to the extensive group of mutations in the spike protein. The first version of the manuscript was generally strong, although some limitations were noted including an incomplete description of the ZCB11 binding epitope and rather limited protection studies in vivo in the hamster. The authors provided a detailed response in their rebuttal, and addressed virtually every concern raised by the Reviewers including the addition of cryo-EM data on the binding site of ZCB11 as well as new post-infection therapeutic studies that confirm its potential as an immunotherapy. The authors should be commended for their additional effort and excellent contribution to the field. I only have a few small comments.

Response: We thank reviewer #1 for the supportive comments.

Minor Comments.

1. The authors show that ZCB11 has interactions with residue F486 (as do other public clonotype antibodies). Although it is not a major contact, this residue could be a liability for ZCB11 against emerging BA.4 and BA.5 sublineage variants (F486V). Can the authors test ZCB11 against pseudoviruses with this mutation (ideally using the BA.4 or BA.5 sequence)? At a minimum, the authors should point out this residue as a possible one of concern in their limitations section in the context of loss of broadly neutralizing activity.

Response: We thank reviewer #1 for the helpful suggestion. During the revision, we have added new data on Omicron BA.2, which is one of the most prevalent strains recently found in Hong Kong, US and other regions and countries. We, however, have not been able to isolate or generate continuously emerging strains including BA.4 and BA.5. We agree that the residue F486V or other emerging mutations might be a concern. We, therefore, have pointed it out as a possible concern in the revised discussion. We say that “During manuscript revision, we have

noticed that the newly emerged Omicron sublineages BA.4 and BA.5 in South Africa contain a F486V mutation, currently accounting for less than 0.01% of total sequences (<https://www.gisaid.org/>). The V486 residue retains the hydrophobic property in the second interface. Moreover, the first interface, relatively stronger one, is unlikely influenced by V486. Future experiments are needed to determine if V486 would cause a major conformational change in RBM leading to the loss of ZCB11 broadly neutralizing activity” (Lines 380-386).

2. Histopathology. In Figures 5l and 6f, the images lack scale bars. Also, in the legends, the authors state representative images are used, but provide no indication as to how many animals were surveyed, which should be included. Finally, does Omicron really cause significant lung pathology in hamsters (e.g., see (Halfmann et al., 2022) as suggested in Figure 5? Why is there an apparent disparity in the lung disease severity of the PBS-treated and Omicron-infected hamsters in Figures 5 and 6. Were these harvested on the same day? There is no description of the histology in the Methods, and the level of detail in text main text or legends describing what findings are different in Figure 6f (ZCB11 versus PBS treatment) is lacking.

Response: We apologized for the poor labels and other issues related to Figures 5 and 6. We have made following changes: 1) thickened the scale bars; 2) revised the legends and included the number of animals; 3) confirmed lung pathology in several control hamsters and added the Supplementary Fig. 7; 4) explained the hamster gender difference in two Omicron-challenge experiments (Lines 313-315); 5) added a section in Method for histology (Lines 660-668); and 6) provided some details in legends describing the findings in Figures 5l and 6f (Lines 286-288 and 311-313).

We have confirmed lung pathology in several control hamsters in Fig. 5l. Omicron infection indeed caused severe pathology in some areas of lungs. We noticed two major differences compared to the results from Halfmann et al., 2022. First, we used a higher dose of 10^5 PFU (vs 10^3 PFU in Halfmann’s paper) Omicron virus for the challenge experiment. Second, we used 6-10-week-old male hamsters for the experiment in Fig. 5l whereas they used 5–6-week-old male hamsters.

For the apparent disparity in the lung disease severity of the PBS-treated and Omicron-infected hamsters between Fig. 5l and Fig.6f, we realized that the gender of hamsters was likely a major cause. Male and female hamsters were used in Fig. 5l and Fig. 6f, respectively. Our results

indicated that female hamsters showed relatively lower viral titers and milder lung pathogenesis in line with a previous study¹. We have indicated the animal gender in revised figure legends.

LITERATURE CITED.

Halfmann, P.J., Iida, S., Iwatsuki-Horimoto, K., Maemura, T., Kiso, M., Scheaffer, S.M., Darling, T.L., Joshi, A., Loeber, S., Singh, G., et al. (2022). SARS-CoV-2 Omicron virus causes attenuated disease in mice and hamsters. *Nature*.

Reference

- 1 Yuan, L. *et al.* Gender associates with both susceptibility to infection and pathogenesis of SARS-CoV-2 in Syrian hamster. *Signal Transduction and Targeted Therapy* **6**, 136, doi:10.1038/s41392-021-00552-0 (2021).

Reviewer #2 (Remarks to the Author):

Response: We thank reviewer #2 for the supportive comments.

Reviewer #3 (Remarks to the Author):

The authors responded appropriately to all my comments on the original manuscript. They performed additional experiments and added a second tissue target.

In regards to histopathology, additional immunohistochemistry targeting SARS-CoV-2 antigen would have been helpful.

Response: We thank reviewer #3 for the supportive comments. We have added the supplementary Fig. 7 on results of immunohistochemistry targeting SARS-CoV-2 NP antigen.

RESPONSE TO FINAL REVIEWER COMMENTS –

Reviewer #1 (Remarks to the Author): Review of Zhou et al This revised manuscript addresses a limitation of many current therapeutic antibodies that are being clinically used against SARS-CoV-2, namely their loss of inhibitory activity against the Omicron variant due to the extensive group of mutations in the spike protein. The first version of the manuscript was generally strong, although some limitations were noted including an incomplete description of the ZCB11 binding epitope and rather limited protection studies in vivo in the hamster.

The authors provided a detailed response in their rebuttal, and addressed virtually every concern raised by the Reviewers including the addition of cryo-EM data on the binding site of ZCB11 as well as new post-infection therapeutic studies that confirm its potential as an immunotherapy. The authors should be commended for their additional effort and excellent contribution to the field. I only have a few small comments.

Response: *We thank reviewer #1 for the supportive comments.*

Minor Comments 1. The authors show that ZCB11 has interactions with residue F486 (as do other public clonotype antibodies). Although it is not a major contact, this residue could be a liability for ZCB11 against emerging BA.4 and BA.5 sublineage variants (F486V). Can the authors test ZCB11 against pseudoviruses with this mutation (ideally using the BA.4 or BA.5 sequence)? At a minimum, the authors should point out this residue as a possible one of concern in their limitations section in the context of loss of broadly neutralizing activity.

Response: *We thank reviewer #1 for the helpful suggestion. During the revision, we have added new data on Omicron BA.2, which is one of the most prevalent strains recently found in Hong Kong, US and other regions and countries. We, however, have not been able to isolate or generate continuously emerging strains including BA.4 and BA.5. We agree that the residue F486V or other emerging mutations might be a concern. We, therefore, have pointed it out as a possible concern in the revised discussion. We say that “During manuscript revision, we have noticed that the newly emerged Omicron sublineages BA.4 and BA.5 in South Africa contain a F486V mutation, currently accounting for less than 0.01% of total sequences (<https://www.gisaid.org/>).*

The V486 residue retains the hydrophobic property in the second interface. Moreover, the first interface, relatively stronger one, is unlikely influenced by V486. Future experiments are needed to determine if V486 would cause a major conformational change in RBM leading to the loss of ZCB11 broadly neutralizing activity”

(Lines 380-386). 2. Histopathology. In Figures 5l and 6f, the images lack scale bars. Also, in the legends, the authors state representative images are used, but provide no indication as to how many animals were surveyed, which should be included. Finally, does Omicron really cause significant lung pathology in hamsters (e.g., see (Halfmann et al., 2022) as suggested in Figure 5? Why is there an apparent disparity in the lung disease severity of the PBS-treated and Omicron-infected hamsters in Figures 5 and 6. Were these harvested on the same day?

There is no description of the histology in the Methods, and the level of detail in text main text or legends describing what findings are different in Figure 6f (ZCB11 versus PBS treatment) is lacking.

Response: We apologized for the poor labels and other issues related to Figures 5 and 6. We have made following changes: 1) thickened the scale bars; 2) revised the legends and included the number of animals; 3) confirmed lung pathology in several control hamsters and added the Supplementary Fig. 7; 4) explained the hamster gender difference in two Omicron-challenge experiments (Lines 313-315); 5) added a section in Method for histology (Lines 660-668); and 6) provided some details in legends describing the findings in Figures 5l and 6f (Lines 286-288 and 311-313). We have confirmed lung pathology in several control hamsters in Fig. 5l. Omicron infection indeed caused severe pathology in some areas of lungs. We noticed two major differences compared to the results from Halfmann et al., 2022.

First, we used a higher dose of 105 PFU (vs 103 PFU in Halfmann's paper) Omicron virus for the challenge experiment. Second, we used 6-10-week-old male hamsters for the experiment in Fig. 5l whereas they used 5-6-week-old male hamsters. For the apparent disparity in the lung disease severity of the PBS-treated and Omicroninfected hamsters between Fig. 5l and Fig.6f, we realized that the gender of hamsters was likely a major cause. Male and female hamsters were used in Fig. 5l and Fig. 6f, respectively.

Our results indicated that female hamsters showed relatively lower viral titers and milder lung pathogenesis in line with a previous study¹. We have indicated the animal gender in revised figure legends.

LITERATURE CITED. Halfmann, P.J., Iida, S., Iwatsuki-Horimoto, K., Maemura, T., Kiso, M., Scheaffer, S.M., Darling, T.L., Joshi, A., Loeber, S., Singh, G., et al. (2022). SARS-CoV-2 Omicron virus causes attenuated disease in mice and hamsters. Nature. Reference 1 Yuan, L. et al. Gender associates with both susceptibility to infection and pathogenesis of SARS-CoV-2 in Syrian hamster. Signal Transduction and Targeted Therapy 6, 136, doi:10.1038/s41392-021-00552-0 (2021).

Reviewer #2 (Remarks to the Author): All of my concerns were addressed. The manuscript looks excellent.

Response: We thank reviewer #2 for the supportive comments.

Reviewer #3 (Remarks to the Author): The authors responded appropriately to all my comments on the original manuscript. They performed additional experiments and added a second tissue target. In regards to histopathology, additional immunohistochemistry targeting SARS-CoV-2 antigen would have been helpful.

Response: We thank reviewer #3 for the supportive comments. We have added the supplementary Fig. 7 on results of immunohistochemistry targeting SARS-CoV-2 NP antigen.